# WaterFlow: Fast and Robust Watermarking in the Latent Fourier Domain

## Abstract

The rise of high-fidelity generative imagery has amplified the need for practical and robust visual watermarking techniques. However, existing methods often suffer from high computational cost and fail to withstand modern generative and adversarial attacks, limiting their real-world applicability. In this work, we present **WaterFlow**, a fast, lightweight, and highly robust watermarking framework that embeds hidden signals with high fidelity. WaterFlow leverages pretrained latent diffusion models to insert watermarks directly in the latent space. Unlike prior approaches, it learns a watermark in the Fourier domain of the latent representation—enhancing robustness while preserving perceptual quality. This design enables efficient and accurate watermark detection, even under challenging compound perturbations. Additionally, WaterFlow supports control over the quality–robustness trade-off without retraining, making it adaptable to diverse use cases. We evaluate WaterFlow on MS-COCO, DiffusionDB, and WikiArt, where it consistently outperforms prior methods in robustness while matching the image quality of top-performing approaches.

## 1 Introduction

Watermarking techniques for digital content have been studied for decades, with early efforts focusing on embedding imperceptible signals into images to assert ownership and authenticity (Cox et al., 2002). As synthetic image generation becomes increasingly accessible and photorealistic, visual watermarking plays a critical role in preventing misuse of generative models—such as creating misleading or unauthorized content (Franceschelli & Musolesi, 2022).

A core challenge in visual watermarking is achieving robustness without sacrificing perceptual quality. The watermark must remain invisible to the human eye and yet reliably detectable, even after common transformations like compression, rotation, or adversarial perturbations. Classical methods that modify texture-rich regions (Bender et al., 1996), frequency domains (Kundur & Hatzinakos, 1998), or low-order bits (Wolfgang & Delp, 1996) often struggle under modern transformations.

Recent deep learning approaches (Zhu et al., 2018; Luo et al., 2020; Zhang et al., 2019b) train end-to-end models to embed and detect robust watermarks. While these methods improve resilience, they are often vulnerable to attacks from modern generative models like VAEs and diffusion models (Zhao et al., 2023a; Ballé et al., 2018; Cheng et al., 2020), which can regenerate clean content and effectively erase embedded signals.

Complementary efforts embed watermarks into the generative process itself—either via dataset poisoning (Zhao et al., 2023b) or by modifying sampling trajectories (Fernandez et al., 2023). Diffusion models, in particular, offer semantically rich latent spaces that are well-suited to watermarking. For instance, Tree-Ring (Wen et al., 2024) encodes frequency-domain signals in latents, while Stable Signature (Fernandez et al., 2023) leverages diffusion sampling to enforce watermark persistence.

Building on these insights, ZoDiac (Zhang et al., 2024) extends watermarking to arbitrary images by optimizing latent-space vectors in Stable Diffusion. Although it yields strong defenses against generative attacks, its per-image optimization loop is slow and it falters under intense combination attacks. By contrast, VINE (Lu et al., 2024) trains diffusion models end-to-end—at the cost of massive training datasets and heavyweight encoder, decoder, and discriminator networks—making it highly computationally intensive.

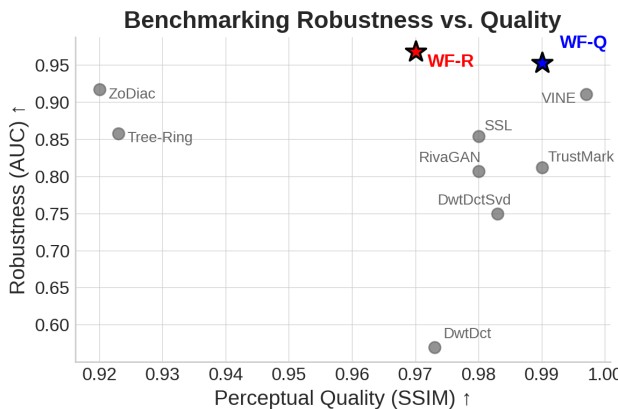

Figure 1: We showcase visualized robustness vs. perceptual results. We average AUC and SSIM across all three dataset. Results in the upper-right corner are preferred. We observe that WF-R has the highest robustness with competitive quality while WF-Q is one of the best in terms of quality and second place in overall robustness.

In this work, we propose **WaterFlow** (WF), a fast, robust, and lightweight visual watermarking method that applies to both real and synthetic images. WaterFlow operates in three phases: training, embedding, and detection. During training, we learn a compact, flow-based generator that produces watermark patterns conditioned on an image's latent representation. To embed a watermark, we extract latents using a pretrained diffusion model, transform them into the frequency domain, and inject a custom watermark before image generation. Frequency-space watermarking has long been associated with robustness (Solachidis & Pitas, 2001), and WaterFlow builds on this principle without sacrificing visual fidelity.

We introduce two variants of WaterFlow: **WaterFlow-Robust** (WF-R), which prioritizes robustness, and **WaterFlow-Quality** (WF-Q), which emphasizes image quality while still maintaining strong robustness. Both variants share the same underlying model, trained once and reused at inference. A simple adjustment to a postprocessing parameter allows users to seamlessly switch between them, enabling real-time adaptability based on downstream requirements.

We evaluated WaterFlow across three diverse datasets—MS-COCO (Lin et al., 2014), DiffusionDB (Wang et al., 2022), and WikiArt (Phillips & Mackintosh, 2011)—and benchmark against several state-of-the-art watermarking methods under a range of adversarial attacks.

**Our contributions are as follows:**

- We introduce WaterFlow, a practical and efficient visual watermarking framework that achieves state-of-the-art general robustness without compromising image quality.
- We propose a learned latent-to-frequency mapping with a new loss term that adaptively balances invisibility and detectability across diverse image types.
- WaterFlow is the *first* watermarking method to withstand complex combination attacks while maintaining top-tier resistance to generative removal.
- We enable a dynamic trade-off between image quality and robustness via a simple postprocessing adjustment, allowing users to adapt the watermarking behavior without retraining.

## 2 RELATED WORK

### 2.1 IMAGE WATERMARKING

Traditional watermarking methods rely on frequency-domain decompositions (e.g., DCT, wavelets, Fourier) for robustness to standard image transformations Bors & Pitas (1996); Xia et al. (1998); Urvoy et al. (2014). More recently, end-to-end DNN-based approaches jointly train encoders and

decoders to balance imperceptibility and robustness Hayes & Danezis (2017); Zhu et al. (2018); Tancik et al. (2019), with extensions using GANs Zhang et al. (2019b;a); Huang et al. (2023); Ma et al. (2022) and invertible networks Ma et al. (2022). While effective against conventional distortions, these methods are vulnerable to generative attacks that regenerate images and erase watermarks Zhao et al. (2023a).

## 2.2 WATERMARKING VIA DIFFUSION MODELS

Recent work has focused on diffusion-based watermarking to withstand such attacks. One line fine-tunes diffusion models to generate watermarked synthetic content Wang et al. (2023); Cui et al. (2023); Zhao et al. (2023b), as in Stable Signature Fernandez et al. (2023) and WaDiff Min et al. (2024), but these approaches cannot watermark natural images. Other methods embed watermarks into latent spaces, such as Tree-Ring Wen et al. (2024), which uses frequency-domain signals retrievable via DDIM inversion, and ZoDiac Zhang et al. (2024), which optimizes latent-space watermarks for robustness but requires slow per-image optimization. Subsequent methods (e.g., VINE Lu et al. (2024)) bridge latent embedding with encoder-decoder architectures by leveraging diffusion models as encoders, but at significant computational cost. These approaches improve resistance to regeneration-based attacks Zhao et al. (2023a), yet often struggle with compound perturbations and geometric transformations.

## 2.3 DIFFUSION MODELS AND DDIM

We briefly summarize the fundamentals of diffusion models, particularly DDIM sampling Ho et al. (2020); Song et al. (2020b); Dhariwal & Nichol (2021). In a forward diffusion process, a clean image $x_0$ is gradually transformed into noise $x_T$ over $T$ time steps:

$$q(x_t|x_{t-1}) = \mathcal{N}(x_t; \sqrt{1-\beta_t}x_{t-1}, \beta_t I), \tag{1}$$

where each step is Markovian, and $\beta_t \in (0,1)$ controls the noise level. The closed-form expression for any step $x_t$ is:

$$x_t = \sqrt{\bar{\alpha}_t}x_0 + \sqrt{1-\bar{\alpha}_t}\epsilon, \tag{2}$$

with $\bar{\alpha}_t = \prod_{s=1}^{t}(1-\beta_s)$. In the reverse process, DDIM Song et al. (2020a) offers a deterministic approach to reconstruct $x_0$ from noise $x_T$ by estimating $\epsilon_\theta(x_t)$, the predicted noise at step $t$:

$$\tilde{x}_0^t = \frac{x_t - \sqrt{1-\bar{\alpha}_t}\epsilon_\theta(x_t)}{\sqrt{\bar{\alpha}_t}}. \tag{3}$$

Then, $x_{t-1}$ is calculated as:

$$x_{t-1} = \sqrt{\bar{\alpha}_{t-1}}\tilde{x}_0^t + \sqrt{1-\bar{\alpha}_{t-1}}\epsilon_\theta(x_t). \tag{4}$$

This recursive process enables generation from $x_T$ to $x_0$:

$$x_0 = \mathcal{G}_\theta(x_T).$$

Moreover, the inverse process—starting from a real image $x_0$—can recover the initial latent $x_T$ using:

$$x_{t+1} = \sqrt{\bar{\alpha}_{t+1}}\tilde{x}_0^t + \sqrt{1-\bar{\alpha}_{t+1}}\epsilon_\theta(x_t). \tag{5}$$

This inversion, denoted $x_T = \mathcal{G}'(x_0)$, maps the image to its latent representation $Z_T$, enabling watermarking directly in the latent space.

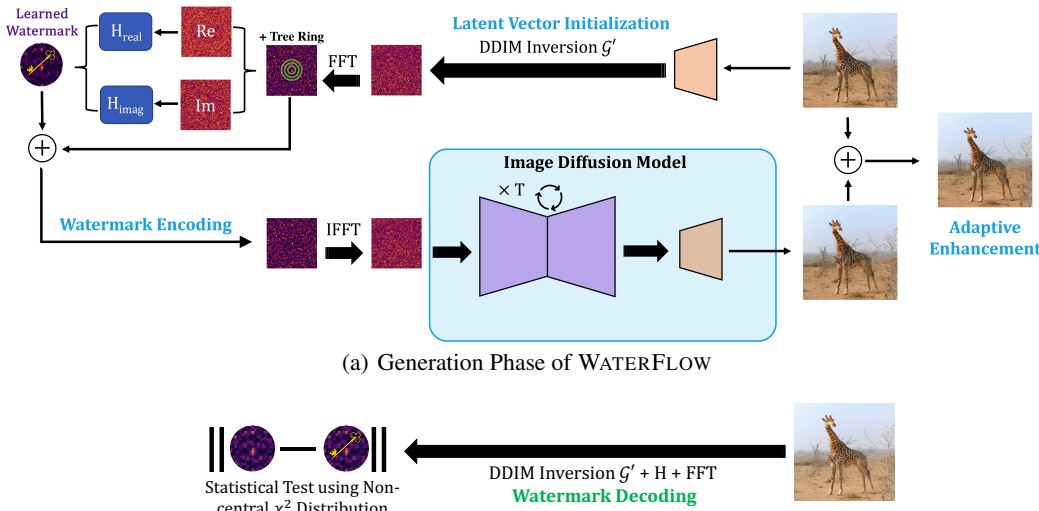

(a) Generation Phase of WATERFLOW

(b) Detection Phase of WATERFLOW

Figure 2: Overview of WATERFLOW. (a) shows the generation phase, while (b) illustrates the detection phase.

## 3 APPROACH

WaterFlow is a zero-shot watermarking framework that leverages invertible transformations and pre-trained diffusion models to enable *robust, fast, and high-fidelity* watermarking. We implant learned watermarks in the Fourier-transformed latent space of images, enabling reliable detection via latent-space recovery. Figure 2 provides an overview.

### 3.1 WATERMARKING PROCEDURE

Algorithm 1 outlines the full watermarking procedure. Given an image $x_0$, we first apply DDIM inversion to obtain the latent representation $Z_T = \mathcal{G}'(x_0)$. We then implant an initial Tree-Ring watermark (Wen et al., 2023) into the Fourier domain of $Z_T$ using a circular binary mask $M$:

$$\mathcal{F}(Z_T') = \mathcal{F}(Z_T) \odot (1 - M) + M \odot W_{\text{Tree}} \tag{6}$$

$W_{\text{Tree}}$ consists of concentric rings sampled from the Fourier transform of the original latent.

Next, we generate an input-specific watermark $W^*$ using two neural networks, $H_{\text{real}}$ and $H_{\text{imag}}$, which operate on the real and imaginary components of $\mathcal{F}(Z_T')$, respectively:

$$W^* = H_{\text{real}}(\Re(\mathcal{F}(Z_T'))) + j \cdot H_{\text{imag}}(\Im(\mathcal{F}(Z_T'))) \tag{7}$$

We implant $W^*$ into the last channel of the latent:

$$M(x, y) = \begin{cases} 1 & \text{if } \sqrt{x^2 + y^2} \leq r \\ 0 & \text{otherwise} \end{cases} \tag{8}$$

$$\mathcal{F}(Z_{W^*}) = \mathcal{F}(Z_T')[-1, :, :] \odot (1 - M) + M \odot W^* \tag{9}$$

We then apply the inverse Fourier transform and decode $Z_{W^*}$ using $\mathcal{G}$ to obtain the watermarked image $\hat{x}_0$. Note that we do not guarantee Hermitian symmetry anymore, meaning we take the real portion from the IFT. To ensure fidelity, we adopt ZoDiac's adaptive enhancement (Zhang et al., 2024), introducing a blending factor $\gamma$ to enforce an SSIM threshold $s^*$:

$$\bar{x}_0 = \gamma x_0 + (1 - \gamma)\hat{x}_0 \tag{10}$$

We search for the minimal $\gamma$ such that $\text{SSIM}(\bar{x}_0, x_0) \geq s^*$. This parameter is what will allow us to effectively serve different versions of WaterFlow without retraining.

---

**Algorithm 1** WaterFlow Watermarking

---

**Require:** Image $x_0$, binary mask $M$, pretrained LDM $\mathcal{G}$ and inversion $\mathcal{G}'$, trained mappings $H_{\text{real}}$, $H_{\text{imag}}$, Tree-Ring watermark $W_{\text{Tree}}$, SSIM threshold $s^*$
1: $Z_T = \mathcal{G}'(x_0)$
2: $\mathcal{F}(Z_T') = \mathcal{F}(Z_T) \odot (1 - M) + M \odot W_{\text{Tree}}$
3: $W^* = H_{\text{real}}(\Re(\mathcal{F}(Z_T'))) + j \cdot H_{\text{imag}}(\Im(\mathcal{F}(Z_T')))$
4: $\mathcal{F}(Z_{W^*}) = \mathcal{F}(Z_T')[-1, :, :] \odot (1 - M) + M \odot W^*$
5: $\hat{x}_0 = \mathcal{G}(Z_{W^*})$
6: Search $\gamma \in [0, 1]$ s.t. $\text{SSIM}(\bar{x}_0, x_0) \geq s^*$
7: $\bar{x}_0 = \gamma x_0 + (1 - \gamma)\hat{x}_0$
8: Output - $\bar{x}_0$

---

## 3.2 TRAINABLE WATERMARK

In contrast to prior works that use fixed or latent-optimized watermarks (Wen et al., 2024; Zhang et al., 2024), we propose a trainable watermark that adapts per-image while modifying only a small latent region. We use lightweight Residual Flow networks (Chen et al., 2019) for $H_{\text{real}}$ and $H_{\text{imag}}$ to map real and imaginary components of the latent's Fourier space.

Our training loss is:

$$\mathcal{L} = \lambda_2 \mathcal{L}_2(x_0, \hat{x}_0) + \lambda_s \mathcal{L}_s(x_0, \hat{x}_0) + \lambda_p \mathcal{L}_p(x_0, \hat{x}_0) + \lambda_n \mathcal{L}_n(Z_T, W^*, M) \tag{11}$$

Here, $\mathcal{L}_2$ is MSE loss, $\mathcal{L}_s$ is SSIM loss (Zhao et al., 2017), $\mathcal{L}_p$ is VGG perceptual loss (Johnson et al., 2016), and $\mathcal{L}_n$ encourages separability between the original latent and learned watermark:

$$\mathcal{L}_n = -\frac{1}{wh} \sum_{i,j} \left([\mathcal{F}(Z_T) \odot M]_{i,j} - [W^* \odot M]_{i,j}\right)^2 \tag{12}$$

## 3.3 WATERMARK DETECTION

To detect a watermark in a given image $x_0$, we project it to latent space and extract the last channel: $y = \mathcal{F}(\mathcal{G}'(x_0))[-1, :, :]$. Under the null hypothesis $H_0$, we assume $y \sim \mathcal{N}(0, \sigma^2 I_{\mathbb{C}})$. We estimate $\sigma^2$ from the masked region:

$$\sigma^2 = \frac{1}{\sum M} \sum (M \odot y)^2$$

We then compute a detection score:

$$\eta = \frac{1}{\sigma^2} \sum (M \odot W^* - M \odot y)^2 \tag{13}$$

$\eta$ follows a non-central chi-squared distribution (PATNAIK, 1949) with degrees of freedom $q = \sum M$ and non-centrality $\lambda = \frac{1}{\sigma^2} \sum (M \odot W^*)^2$.

The p-value for detection is:

$$p = \Pr(\chi_{q,\lambda}^2 \leq \eta \mid H_0)$$

We define the detection probability as $1 - p$, where low $p$ indicates strong watermark presence.

# 4 EXPERIMENTS

In this section, we detail the datasets, settings used by our approach, the baselines used for comparison, robustness, and runtime performance.

## 4.1 SET UP

We evaluate our approach on three distinct datasets that span a comprehensive set of images.

**MS-COCO** (Lin et al., 2014) provides real-world images randomly sampled from a large-scale benchmark commonly used for image recognition and segmentation. The dataset features a wide range of everyday scenes, including natural landscapes, people, animals, food, vehicles, and objects.

Table 1: AUC between the non-watermarked and the watermarked image evaluated under a series of attacks. We group results by dataset and bold the top three average AUCs per dataset (best in gold, second in silver, third in bronze).

| Dataset | Method | Post-Attack | | | | | | | | | | All | All w/o Rotation | Overall Avg. |
| | | Brightness | Contrast | JPEG | Rotation | G-Noise | G-Blur | BM3D | Bmshj18 | Cheng20 | Zhao23 | | | |
|---|---|---|---|---|---|---|---|---|---|---|---|---|---|---|
| MS-COCO | DwtDct | 0.498 | 0.505 | 0.498 | 0.564 | 0.501 | 0.505 | 0.904 | 0.816 | 0.535 | 0.417 | 0.503 | 0.494 | 0.562 |
| | DwtDctSvd | 0.491 | 0.492 | 0.991 | 0.682 | 1.000 | 1.000 | 0.955 | 0.799 | 0.796 | 0.811 | 0.489 | 0.507 | 0.751 |
| | RivaGan | 1.000 | 1.000 | 1.000 | 0.455 | 1.000 | 1.000 | 0.999 | 0.718 | 0.716 | 0.831 | 0.494 | 0.529 | 0.812 |
| | SSL | 1.000 | 1.000 | 0.994 | 1.000 | 0.984 | 1.000 | 0.919 | 0.744 | 0.758 | 0.922 | 0.507 | 0.502 | 0.861 |
| | Trustmark | 0.997 | 0.995 | 0.987 | 0.501 | 0.992 | 1.000 | 0.998 | 0.920 | 0.829 | 0.488 | 0.511 | 0.501 | 0.810 |
| | VINE | 1.000 | 1.000 | 1.000 | 0.508 | 1.000 | 1.000 | 1.000 | 1.000 | 1.000 | 1.000 | 0.492 | 0.907 | 0.909 |
| | Tree-Ring | 0.930 | 0.928 | 0.872 | 0.662 | 0.873 | 0.922 | 0.898 | 0.854 | 0.860 | 0.859 | 0.575 | 0.706 | 0.828 |
| | ZoDiac | 0.996 | 0.996 | 0.989 | 0.771 | 0.987 | 0.996 | 0.994 | 0.980 | 0.979 | 0.978 | 0.618 | 0.809 | **0.924** |
| | **WF-R (Ours)** | 0.997 | 0.997 | 0.996 | 0.857 | 0.995 | 0.998 | 0.998 | 0.996 | 0.997 | 0.994 | 0.863 | 0.977 | **0.972** |
| | **WF-Q (Ours)** | 0.986 | 0.987 | 0.985 | 0.842 | 0.986 | 0.990 | 0.986 | 0.982 | 0.985 | 0.978 | 0.800 | 0.935 | **0.954** |
| DiffusionDB | DwtDct | 0.509 | 0.511 | 0.551 | 0.433 | 0.878 | 0.790 | 0.536 | 0.509 | 0.513 | 0.511 | 0.504 | 0.504 | 0.563 |
| | DwtDctSvd | 0.431 | 0.433 | 0.979 | 0.725 | 0.989 | 1.000 | 0.960 | 0.813 | 0.812 | 0.742 | 0.493 | 0.504 | 0.740 |
| | RivaGan | 0.996 | 0.997 | 0.993 | 0.499 | 0.999 | 1.000 | 0.991 | 0.697 | 0.672 | 0.721 | 0.496 | 0.538 | 0.800 |
| | SSL | 1.000 | 0.999 | 0.970 | 0.999 | 0.963 | 1.000 | 0.882 | 0.716 | 0.717 | 0.824 | 0.514 | 0.527 | 0.843 |
| | Trustmark | 1.000 | 0.998 | 0.989 | 0.498 | 0.995 | 1.000 | 0.999 | 0.945 | 0.860 | 0.504 | 0.495 | 0.504 | 0.816 |
| | VINE | 1.000 | 1.000 | 1.000 | 0.495 | 1.000 | 1.000 | 1.000 | 1.000 | 1.000 | 1.000 | 0.496 | 0.882 | 0.906 |
| | Tree-Ring | 0.953 | 0.947 | 0.923 | 0.669 | 0.924 | 0.954 | 0.940 | 0.889 | 0.914 | 0.886 | 0.584 | 0.758 | 0.862 |
| | ZoDiac | 0.990 | 0.989 | 0.980 | 0.773 | 0.982 | 0.992 | 0.990 | 0.972 | 0.973 | 0.959 | 0.618 | 0.838 | **0.921** |
| | **WF-R (Ours)** | 0.994 | 0.996 | 0.996 | 0.996 | 0.997 | 0.996 | 0.997 | 0.999 | 0.693 | 0.997 | 0.713 | 0.978 | **0.946** |
| | **WF-Q (Ours)** | 0.991 | 0.990 | 0.991 | 0.680 | 0.991 | 0.996 | 0.994 | 0.989 | 0.990 | 0.980 | 0.661 | 0.945 | **0.933** |
| WikiArt | DwtDct | 0.503 | 0.506 | 0.547 | 0.423 | 0.896 | 0.809 | 0.547 | 0.499 | 0.501 | 0.488 | 0.515 | 0.497 | 0.561 |
| | DwtDctSvd | 0.505 | 0.509 | 0.985 | 0.721 | 1.000 | 1.000 | 0.967 | 0.819 | 0.828 | 0.753 | 0.510 | 0.495 | 0.758 |
| | RivaGan | 0.997 | 0.997 | 0.995 | 0.466 | 0.996 | 0.997 | 0.992 | 0.735 | 0.701 | 0.819 | 0.500 | 0.527 | 0.810 |
| | SSL | 1.000 | 1.000 | 0.988 | 1.000 | 0.982 | 1.000 | 0.897 | 0.737 | 0.750 | 0.883 | 0.538 | 0.515 | 0.857 |
| | Trustmark | 0.995 | 0.992 | 0.980 | 0.504 | 0.995 | 0.999 | 0.997 | 0.920 | 0.816 | 0.504 | 0.499 | 0.502 | 0.809 |
| | VINE | 1.000 | 1.000 | 1.000 | 0.503 | 1.000 | 1.000 | 1.000 | 1.000 | 0.503 | 1.000 | 0.515 | 0.854 | 0.906 |
| | Tree-Ring | 0.953 | 0.952 | 0.909 | 0.692 | 0.913 | 0.936 | 0.915 | 0.879 | 0.886 | 0.895 | 0.605 | 0.752 | 0.857 |
| | ZoDiac | 0.992 | 0.992 | 0.981 | 0.732 | 0.983 | 0.992 | 0.987 | 0.964 | 0.962 | 0.959 | 0.628 | 0.801 | **0.915** |
| | **WF-R (Ours)** | 0.998 | 1.000 | 0.999 | 0.925 | 0.999 | 1.000 | 0.999 | 0.998 | 0.999 | 0.999 | 0.909 | 0.990 | **0.985** |
| | **WF-Q (Ours)** | 0.995 | 0.996 | 0.994 | 0.901 | 0.995 | 0.997 | 0.995 | 0.993 | 0.991 | 0.990 | 0.851 | 0.961 | **0.971** |

**DiffusionDB** (Wang et al., 2022) contains images generated by users interacting with Stable Diffusion models. These samples cover a wide semantic and aesthetic range—such as photorealistic scenes, fantasy imagery, cartoons, anime, space-style, surrealism, and stylized portraits—resulting from diverse prompts and generation parameters.

**WikiArt** (Phillips & Mackintosh, 2011) contributes 500 images from a curated collection of artworks across artistic styles and historical periods. The dataset includes oil paintings, sketches, impressionist pieces, abstract art, and other stylized compositions. These images are typically non-photorealistic and feature unique visual traits such as brush textures, symbolic structures, and unconventional color palettes.

For baseline comparisons, we evaluated against *8 key methods*: ZoDiac (Zhang et al., 2024), Tree-Ring (Wen et al., 2023), DwtDct (Cox et al., 2007), and DwtDctSvd (Navas et al., 2008), RivaGan (Zhang et al., 2019c), SSL (Fernandez et al., 2023), VINE (Lu et al., 2024), and Trustmark (Bui et al., 2023) on the same set of images.

## 4.2 WATERMARKING ATTACKS

To benchmark the robustness of our watermarking method, we evaluate its performance under common data augmentations and perturbations. We utilize the following attacks in our assessment: **Brightness** and **Contrast**: with a factor of 0.5, **JPEG**: compression with a quality setting of 50, **Rotation**: by 90 degrees, **G-Noise**: Addition of Gaussian noise with std of 0.05, **G-Blur**: Gaussian blur with kernel size 5 and std 1, **BM3D**: Denoising algorithm with a std of 0.1, **Bmshj18** and **Cheng20**: Two Variational AutoEncoder (VAE) based image compression models, both with compression factors of 3 (Ballé et al., 2018; Cheng et al., 2020), **Zhao23**: Stable diffusion-based image regeneration model, with 60 denoising steps (Zhao et al., 2023a), **All**: Combination of all the attacks, and **All w/o Rotation**: Combination of all the attacks without rotation. These attacks cover a holistic set of the current literature. In the Appendix, we include some additional attacks found in WAVES (An et al., 2024).

**Image Quality:** We calculate the Peak Signal-to-Noise Ratio (PSNR) between the watermarked image, Structural Similarity Index (SSIM) (Wang et al., 2004), and Learned Perceptual Image Patch Similarity (LPIPS) metric (Zhang et al., 2018).

**Robustness:** For measuring the watermark robustness, we report average Watermark Detection Rate (WDR). Given the returned p-value of an image, we consider an image watermarked if the detection probability is greater than some threshold $p^*$. In our experiments we use $p^* = 0.9$ for WaterFlow, Tree-Ring, and ZoDiac. We change the threshold for the rest of the baselines as detailed in Appendix. We also report the Area under the curve (AUC) along with the TPR@1%FPR or the true positive rate given we want $1\%$ false positive rate.

**Time Efficiency:** We measure the average time needed to watermark a single image.

Table 2: Image quality experiments showing detection probability, SSIM, LPIPS, and PSNR . We highlight Det. Prob. above 0.95, SSIM above 0.95, LPIPS below 0.1, and PSNR above 30 as examples of excellent quality. We also include the number of trainable parameters.

| Method | # Trainable Params | MS-COCO | | | | DiffusionDB | | | | WikiArt | | | |
|---|---|---|---|---|---|---|---|---|---|---|---|---|---|
| | | Det. Prob ↑ | SSIM ↑ | LPIPS ↓ | PSNR ↑ | Det. Prob ↑ | SSIM ↑ | LPIPS ↓ | PSNR ↑ | Det. Prob ↑ | SSIM ↑ | LPIPS ↓ | PSNR ↑ |
| DwtDct | 0 | 0.87 | 0.98 | 0.02 | 39.96 | 0.84 | 0.97 | 0.02 | 38.21 | 0.83 | 0.97 | 0.02 | 39.04 |
| DwtDctSvd | 0 | 1.00 | 0.99 | 0.02 | 39.88 | 1.00 | 0.98 | 0.02 | 38.22 | 1.00 | 0.98 | 0.02 | 38.98 |
| RivaGAN | $10^5$ | 1.00 | 0.98 | 0.04 | 40.49 | 0.98 | 0.98 | 0.04 | 40.52 | 0.99 | 0.98 | 0.05 | 40.41 |
| SSL | $10^7$ | 1.00 | 0.98 | 0.07 | 41.79 | 1.00 | 0.98 | 0.06 | 41.85 | 1.00 | 0.98 | 0.07 | 41.78 |
| TrustMark | $10^7$ | 1.00 | 0.99 | 0.01 | 41.96 | 1.00 | 0.99 | 0.01 | 42.08 | 1.00 | 1.00 | 0.01 | 42.27 |
| VINE | $10^9$ | 1.00 | 1.00 | 0.01 | 39.62 | 1.00 | 0.99 | 0.01 | 39.47 | 1.00 | 1.00 | 0.01 | 45.36 |
| Tree-Ring | 0 | 0.93 | 0.92 | 0.11 | 25.63 | 0.95 | 0.92 | 0.09 | 25.70 | 0.94 | 0.92 | 0.13 | 26.51 |
| ZoDiac | $10^4$ | 1.00 | 0.92 | 0.09 | 28.48 | 0.99 | 0.92 | 0.07 | 28.29 | 0.99 | 0.92 | 0.10 | 29.60 |
| **WF-R (Ours)** | $10^4$ | 1.00 | 0.97 | 0.06 | 29.78 | 0.99 | 0.97 | 0.06 | 30.19 | 1.00 | 0.97 | 0.06 | 29.38 |
| **WF-Q (Ours)** | $10^4$ | 0.99 | 0.99 | 0.01 | 35.51 | 0.98 | 0.99 | 0.03 | 35.85 | 0.99 | 0.99 | 0.03 | 34.62 |

### 4.3 ROBUSTNESS RESULTS AND DISCUSSION

Robustness results across AUC, TPR@1%FPR, and WDR are summarized in Table 1 and further in the Appendix. Across all datasets and attacks, *WF-R achieves the strongest general performance on all metrics*, consistently outperforming both classical and recent SOTA baselines. Despite prioritizing image quality, WF-Q ranks *second on all robustness metrics*, confirming that high perceptual fidelity can coexist with robustness.

Under standard perturbations (e.g., brightness, contrast, Gaussian noise, blur), both variants maintain near-perfect detection rates. Geometric attacks, particularly rotation, remain a challenge due to spatial misalignment (Dong et al., 2005). Although not trained with any rotation-specific augmentations, WF-R still achieves >0.85 AUC under rotation—second only to SSL, which was explicitly trained for it—demonstrating strong generalization. Compared to the next best method, **WF-R is more than 10% better** on AUC and TPR for DiffusionDB and WikiArt.

On generative-model-based attacks (Bmshj18, Cheng20, Zhao23), legacy methods like DwtDct, RivaGAN, and SSL degrade significantly. In contrast, WaterFlow's variants maintain strong near-perfect detection, on par with Tree-Ring, ZoDiac, and VINE. We attribute this to DDIM inversion aligning watermark perturbations with latent structure, a property shared by Tree-Ring and ZoDiac. In the most difficult composite scenarios—*All* and *All w/o Rotation*—WF-R is miles ahead of the closest baseline **achieving improvements up to nearly 700% on TPR@1% FPR and 44% on AUC** compared to the closest baseline. We note that all other methods have near zero-detection capability. WaterFlow-Q trails closely, still outperforming all baselines. These results establish WaterFlow as the first watermarking method to withstand aggressive, multi-step perturbations.

This robustness is due to $\mathcal{L}_n$ (Eq. 12), which maximizes the separation between the learned watermark $\mathcal{F}(x_0) \odot M$ and its target $W^* \odot M$. Unlike Tree-Ring's fixed zero-patch approach (Wen et al., 2023), our formulation learns per-latent patches that balance fidelity and resilience.

### 4.4 WATERMARKING SPEED RESULTS AND DISCUSSION

We report the time required to watermark a single image using diffusion-based latent watermarking approaches with the same backbone model for an apt comparison. As shown in Figure 3, WaterFlow

completes watermarking in 6.15 seconds on average—comparable to Tree-Ring (5.77s), and nearly **90 times faster** than ZoDiac (517.68s).

This significant speed advantage comes from WaterFlow's architecture: instead of performing iterative latent optimization per image like ZoDiac, our method uses a universal forward pass through a lightweight model which addds minimal overhead in terms of time. While Tree-Ring is slightly faster, it is relatively the same as WaterFlow. On the other hand, WaterFlow balances speed, robustness, and quality, making it far more practical for real-world use.

While some older methods embed watermarks more quickly, we believe WaterFlow has significant potential for speed improvements. Recent models like SDXL-Turbo (Sauer et al., 2024) achieve extremely fast inference by reducing the number of latent diffusion steps—a strategy fully compatible with our framework. Incorporating such models presents a natural path to substantially accelerating WaterFlow for real-world deployment.

### 4.5 Image Quality Results and Discussion

Table 2 summarizes our perceptual fidelity results (visual examples in Appendix). Both WF-R and WF-Q surpass Tree-Ring and ZoDiac on all three metrics—PSNR, SSIM, and LPIPS. In particular, WF-Q posts among the highest SSIM and LPIPS scores, with PSNR only marginally below the prior state of the art, while WF-R delivers markedly better overall fidelity than either baseline. We also emphasize that SSIM and LPIPS provide more reliable assessments of perceptual quality, since they remain stable under small, imperceptible pixel variations that disproportionately impact PSNR.

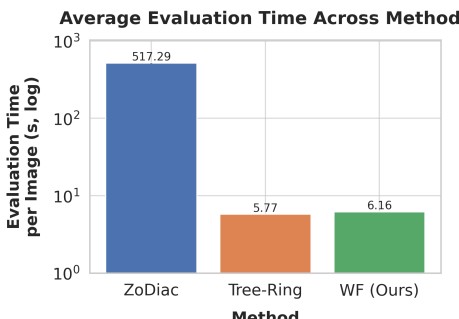

Figure 3: We evaluate the average time it takes to watermark a single image. We evaluate against two methods that share the same architectural backbone. Our results show that against existing strong baselines like ZoDiac, WF is significantly faster.

The adaptive enhancement procedure further boosts image quality with minimal impact on detectability, effectively treating enhancement as a benign perturbation. Unlike methods like VINE and TrustMark that rely on fine-tuning large models with many trainable parameters on a lot of data, WaterFlow achieves competitive or better quality using a much lighter weight model, with many fewer parameters, significantly less data, and much less compute.

In the Appendix, we explore the quality–robustness trade-off more deeply. An ablation study shows that adjusting the SSIM threshold during enhancement can improve perceptual quality with only minor losses in robustness. This flexibility—enabled by a single trained model—highlights WaterFlow's practical value in diverse deployment scenarios.

## 5 Ablation Studies

We perform four key ablations in the main paper, with additional ablations detailed in the Appendix.

### 5.1 Effect of Loss Weight $\lambda_n$

To explore the trade-off between image fidelity and watermark detectability, we vary the robustness loss weight $\lambda_n$ from $10^{-2}$ to $10^{-6}$. As illustrated in Figure 4 (see Appendix for full curves), increasing $\lambda_n$ boosts detection performance at the expense of perceptual quality. This confirms that stronger emphasis on watermark separation enhances detectability but introduces greater distortion.

### 5.2 Choice of Mapping Architecture

We evaluate several architectures, including UNet, MLP, and our proposed FlowNets (Figure 6; detailed in Appendix). While all achieve comparable image quality, UNet and MLP exhibit signifi-

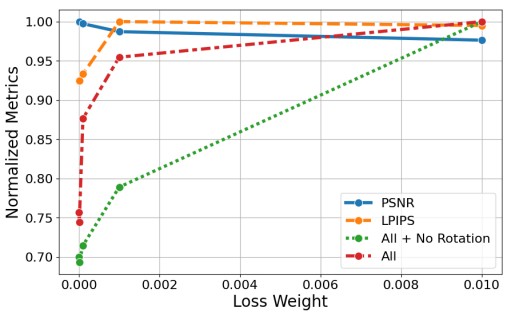

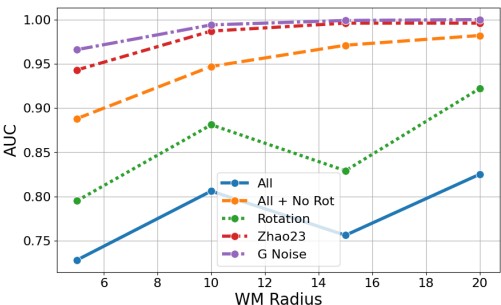

Figure 4: Comparison of loss weights $\lambda_n \in \{10^{-2}, 10^{-3}, 10^{-4}, 10^{-5}, 10^{-6}\}$ across PSNR, LPIPS, and robustness under rotation and combined attacks. PSNR and LPIPS are normalized.

Figure 5: Effect of watermark radius (5, 10, 15, 20) on AUC across multiple attacks. Larger radii is shown to improve robustness across all classes of attacks.

cantly weaker robustness. UNet produces overly weak watermarks that are easily removed by most attacks, whereas MLP withstands basic perturbations but fails against complex attack combinations.

We favor Residual Flows for several reasons. First, their inherent invertibility enables learning more expressive and diverse watermarks without collapse. In contrast, the dimensionality reduction in UNet and MLP can trap the model in poor localizations. Second, residual connections allow the model to default to the already strong Tree-Ring baseline when necessary. Finally, FlowNets are suited to transforming Gaussian latents rather than image pixels, making them a principled choice.

### 5.3 WATERMARK RADIUS

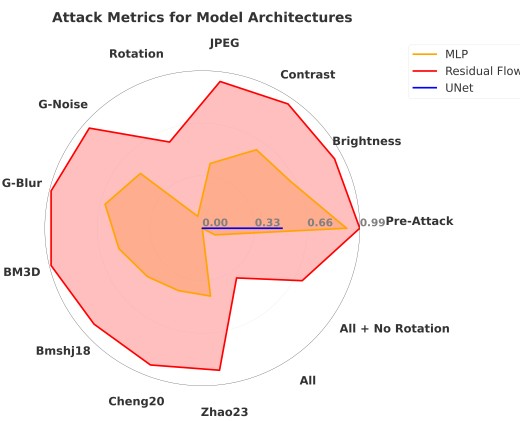

Figure 6: Graph of different model architectures used to parameterize $H_{\text{real}}$ and $H_{\text{imag}}$ as well as their robustness.

We study the impact of the watermark's spatial radius by sweeping values from 5 to 20 pixels (see Appendix for more details). Figure **??** shows that larger radii increase coverage and robustness—e.g., PSNR decreases from 25.49 at radius 5 to 25.10 at radius 20—reflecting the expected fidelity–coverage trade-off.

### 5.4 CROSS-DOMAIN GENERALIZATION

Finally, we train a single WaterFlow-General model jointly on MS-COCO, DiffusionDB, and WikiArt (100 images each). As shown in Table 14 and detailed further in Appendix, this unified model offers slight improvements in perceptual metrics and achieves robustness comparable to ZoDiac, though it remains marginally less robust than our domain-specific variants. These results highlight the strong cross-domain generalization of WaterFlow-General, while also suggesting that multiple lightweight, domain-specialized models can deliver optimal robustness.

## 6 CONCLUSION

WaterFlow is a fast, diffusion-based watermarking framework that combines strong robustness with high visual fidelity. It embeds adaptive watermarks in the latent space of a pretrained diffusion model and supports two modes: WaterFlow-Robust, which sets a new benchmark on DiffusionDB, MS-COCO, and WikiArt while withstanding complex attacks in under one second, and WaterFlow-Quality, which matches or exceeds top baselines in visual quality while maintaining higher detection rates under adversarial attacks.

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

# A    APPENDIX

# B    EXPERIMENT DETAILS

## B.1    DATASET

Our training set for our main experiment consists of 1000 samples. All of our datasets are pulled from publically available huggingface APIs and evaluate on 500 test images. For our ablations we use 300 samples per dataset and evaluate on 100 images for practicality.

## B.2    HARDWARE

All of our training is done on a single NVIDIA A6000 and A5000 GPUs and Intel(R) Xeon(R) Gold 6342 CPU @ 2.80GHz. Our training was completed in around a couple of hours on a singular GPU.

## B.3    HYPERPARAMETERS

We train for a maximum of 5 epochs, Adam optimizer with $\beta_1 = 0.9$ and $\beta_2 = 0.999$, and learning rate 0.001. All our experiments use a batch size of 2. We use 50 denoising steps for our diffusion model. We also use a SSIM threshold of 0.97 for adaptive enhancement for WF-R and 0.99 for WF-Q.

For the loss function, we use $\lambda_n = 10^{-2}, \lambda_2 = 10.0, \lambda_s = 0.1, \lambda_p = 1.0$.

## B.4    CHECKPOINT SELECTION

We take the checkpoint that has a good tradeoff of perceptual quality and L2 loss. We find that an L2 in the hundreds is required for good robustness.

## B.5    BASELINES

For ZoDiac, we adhere to the parameter settings specified in the original manuscript, including an SSIM threshold of 0.92 and 100 epochs. For Tree-Ring, we adapt the original method—designed for watermarking only diffusion-generated images—to support watermarking arbitrary images. This adaptation involves performing DDIM inversion on an input image, embedding the Tree-Ring watermark into the latent space, and then regenerating the image. We also use adaptive enhancement with Tree-Rings. For DwtDct, DwtDctSvd, SSL, and RivaGAN, we embed a 32-bit message and set a watermark detection threshold of 24/32 correctly predicted bits. Each of these methods is executed using its default parameters as provided in their respective implementations. For TrustMark, we use a detection threshold of 41/61 and 64/100 for VINE.

# C    ADDITIONAL RESULTS

## C.1    MAIN EXPERIMENT

We present Table 4 for comprehensive results. We show the TPR 1%FPR along with the AUC. This new metric gives us a sense about how well our detector is given a specially chosen false positive threshold.

## C.2    LOSS WEIGHT ABLATION

We present full results in Table 6, Table 5, Table 7, and Table 8. We use the hyperparameters listed in Appendix A.3 except that we use a SSIM threshold of 0.92.

Our results indicate a general trade-off between perceptual quality and detectability/robustness with the loss weight. That is higher loss weights have lower perceptual quality (PSNR, LPIPS) but are better in robustness metrics (for AUC, TPR1%FPR, WDR).

Table 3: We report the Watermark Detection Rate (WDR) following various perturbations applied to watermarked images. The attacks are grouped as follows: traditional perturbations (Brightness through BM3D), regenerative attacks (Bmshj18 through Zhao23), and combination attacks (All, All w/o Rotation). We bold the highest averaged WDR per dataset.

| Dataset | Method | Brightness | Contrast | JPEG | Rotation | G-Noise | G-Blur | BM3D | Bmshj18 | Cheng20 | Zhao23 | All | All w/o Rotation | Overall Avg. |
|---|---|---|---|---|---|---|---|---|---|---|---|---|---|---|
| | | | | | | | Post-Attack | | | | | | | |
| MS-COCO | DwtDct | 0.000 | 0.000 | 0.004 | 0.000 | 0.626 | 0.306 | 0.000 | 0.000 | 0.000 | 0.000 | 0.000 | 0.000 | 0.078 |
| | DwtDctSvd | 0.106 | 0.104 | 0.782 | 0.000 | 0.998 | 1.000 | 0.474 | 0.026 | 0.020 | 0.130 | 0.000 | 0.000 | 0.303 |
| | RivaGAN | 0.996 | 0.998 | 0.984 | 0.000 | 1.000 | 0.998 | 0.972 | 0.014 | 0.008 | 0.098 | 0.000 | 0.000 | 0.506 |
| | SSL | 1.000 | 1.000 | 0.740 | 0.998 | 0.628 | 1.000 | 0.168 | 0.012 | 0.034 | 0.160 | 0.002 | 0.000 | 0.479 |
| | TrustMark | 0.994 | 0.990 | 0.974 | 0.020 | 0.984 | 1.000 | 0.996 | 0.846 | 0.668 | 0.004 | 0.034 | 0.030 | 0.628 |
| | VINE | 1.000 | 1.000 | 1.000 | 0.078 | 1.000 | 1.000 | 1.000 | 1.000 | 1.000 | 1.000 | 0.074 | 0.716 | 0.822 |
| | Tree-Ring | 0.736 | 0.746 | 0.610 | 0.260 | 0.580 | 0.718 | 0.630 | 0.536 | 0.530 | 0.000 | 0.138 | 0.260 | 0.581 |
| | ZoDiac | 0.982 | 0.986 | 0.960 | 0.434 | 0.954 | 0.986 | 0.980 | 0.920 | 0.920 | 0.904 | 0.180 | 0.434 | 0.803 |
| | **WF-R (Ours)** | 0.974 | 0.976 | 0.978 | 0.988 | 0.990 | 0.986 | 0.974 | 0.992 | 0.524 | 0.982 | 0.542 | 0.916 | **0.902** |
| | WF-Q (Ours) | 0.953 | 0.955 | 0.950 | 0.476 | 0.943 | 0.964 | 0.953 | 0.945 | 0.936 | 0.911 | 0.379 | 0.784 | 0.846 |
| DIFFDB | DwtDct | 0.000 | 0.000 | 0.002 | 0.002 | 0.526 | 0.304 | 0.002 | 0.000 | 0.000 | 0.000 | 0.000 | 0.000 | 0.070 |
| | DwtDctSvd | 0.094 | 0.090 | 0.824 | 0.000 | 0.974 | 1.000 | 0.606 | 0.038 | 0.042 | 0.110 | 0.000 | 0.000 | 0.315 |
| | RivaGAN | 0.948 | 0.944 | 0.898 | 0.000 | 0.958 | 0.970 | 0.878 | 0.004 | 0.004 | 0.120 | 0.002 | 0.002 | 0.477 |
| | SSL | 0.986 | 0.994 | 0.588 | 0.978 | 0.564 | 0.996 | 0.138 | 0.020 | 0.030 | 0.102 | 0.000 | 0.000 | 0.450 |
| | TrustMark | 1.000 | 0.996 | 0.978 | 0.028 | 0.990 | 1.000 | 0.998 | 0.892 | 0.732 | 0.020 | 0.020 | 0.036 | 0.641 |
| | VINE | 1.000 | 1.000 | 1.000 | 0.066 | 1.000 | 1.000 | 1.000 | 1.000 | 1.000 | 1.000 | 0.062 | 0.688 | 0.818 |
| | Tree-Ring | 0.808 | 0.812 | 0.742 | 0.262 | 0.720 | 0.814 | 0.766 | 0.648 | 0.620 | 0.000 | 0.128 | 0.300 | 0.682 |
| | ZoDiac | 0.964 | 0.968 | 0.938 | 0.440 | 0.932 | 0.972 | 0.960 | 0.880 | 0.898 | 0.844 | 0.178 | 0.504 | 0.775 |
| | **WF-R (Ours)** | 0.986 | 0.986 | 0.982 | 0.136 | 0.986 | 0.990 | 0.990 | 0.978 | 0.980 | 0.960 | 0.080 | 0.916 | **0.831** |
| | WF-Q (Ours) | 0.922 | 0.930 | 0.926 | 0.108 | 0.928 | 0.950 | 0.942 | 0.912 | 0.906 | 0.852 | 0.042 | 0.688 | 0.759 |
| WIKIART | DwtDct | 0.000 | 0.000 | 0.002 | 0.000 | 0.586 | 0.274 | 0.000 | 0.000 | 0.000 | 0.000 | 0.000 | 0.000 | 0.072 |
| | DwtDctSvd | 0.080 | 0.080 | 0.804 | 0.000 | 1.000 | 1.000 | 0.636 | 0.046 | 0.080 | 0.150 | 0.000 | 0.000 | 0.323 |
| | RivaGAN | 0.980 | 0.980 | 0.976 | 0.000 | 0.984 | 0.986 | 0.972 | 0.014 | 0.008 | 0.140 | 0.000 | 0.000 | 0.503 |
| | SSL | 1.000 | 1.000 | 0.764 | 0.996 | 0.696 | 1.000 | 0.176 | 0.024 | 0.034 | 0.112 | 0.000 | 0.000 | 0.484 |
| | TrustMark | 0.990 | 0.984 | 0.962 | 0.028 | 0.990 | 0.998 | 0.994 | 0.846 | 0.646 | 0.030 | 0.028 | 0.032 | 0.627 |
| | VINE | 1.000 | 1.000 | 1.000 | 0.074 | 1.000 | 1.000 | 1.000 | 1.000 | 1.000 | 1.000 | 0.058 | 0.598 | 0.811 |
| | Tree-Ring | 0.822 | 0.832 | 0.710 | 0.264 | 0.682 | 0.762 | 0.688 | 0.602 | 0.590 | 0.000 | 0.148 | 0.288 | 0.668 |
| | ZoDiac | 0.962 | 0.960 | 0.906 | 0.350 | 0.918 | 0.960 | 0.944 | 0.830 | 0.838 | 0.798 | 0.170 | 0.418 | 0.755 |
| | **WF-R (Ours)** | 0.994 | 0.998 | 0.988 | 0.668 | 0.994 | 0.996 | 0.990 | 0.984 | 0.986 | 0.982 | 0.550 | 0.950 | **0.923** |
| | WF-Q (Ours) | 0.956 | 0.964 | 0.964 | 0.600 | 0.972 | 0.970 | 0.968 | 0.954 | 0.956 | 0.934 | 0.356 | 0.836 | 0.869 |

Table 4: TPR@1%FPR after the watermarked images undergo a series of perturbations or attacks across MS-COCO, DiffusionDB, and WikiArt. We bold the highest average TPR per dataset.

| Dataset | Method | Brightness | Contrast | JPEG | Rotation | G-Noise | G-Blur | BM3D | Bmshj18 | Cheng20 | Zhao23 | All | All w/o Rotation | Overall Avg. |
|---|---|---|---|---|---|---|---|---|---|---|---|---|---|---|
| | | | | | | | Post-Attack | | | | | | | |
| MS-COCO | DwtDct | 0.012 | 0.000 | 0.000 | 0.014 | 0.000 | 0.000 | 0.740 | 0.400 | 0.004 | 0.006 | 0.012 | 0.000 | 0.096 |
| | DwtDctSvd | 0.172 | 0.176 | 0.922 | 0.000 | 0.998 | 1.000 | 0.630 | 0.104 | 0.102 | 0.248 | 0.006 | 0.012 | 0.364 |
| | RivaGan | 0.996 | 0.998 | 0.994 | 0.004 | 1.000 | 1.000 | 0.992 | 0.072 | 0.080 | 0.272 | 0.014 | 0.010 | 0.536 |
| | SSL | 1.000 | 1.000 | 0.852 | 1.000 | 0.828 | 1.000 | 0.276 | 0.086 | 0.062 | 0.398 | 0.004 | 0.016 | 0.543 |
| | Trustmark | 0.994 | 0.990 | 0.974 | 0.008 | 0.984 | 1.000 | 0.996 | 0.846 | 0.668 | 0.002 | 0.022 | 0.016 | 0.625 |
| | VINE | 1.000 | 1.000 | 1.000 | 0.004 | 1.000 | 1.000 | 1.000 | 1.000 | 1.000 | 1.000 | 0.008 | 0.428 | 0.787 |
| | Tree-Ring | 0.464 | 0.454 | 0.326 | 0.110 | 0.422 | 0.454 | 0.432 | 0.332 | 0.316 | 0.298 | 0.046 | 0.104 | 0.350 |
| | ZoDiac | 0.958 | 0.954 | 0.918 | 0.224 | 0.888 | 0.964 | 0.926 | 0.856 | 0.860 | 0.700 | 0.090 | 0.270 | 0.717 |
| | **WF-R (Ours)** | 0.944 | 0.954 | 0.942 | 0.970 | 0.964 | 0.970 | 0.958 | 0.972 | 0.384 | 0.956 | 0.412 | 0.868 | **0.858** |
| | WF-Q (Ours) | 0.932 | 0.934 | 0.923 | 0.347 | 0.923 | 0.934 | 0.902 | 0.889 | 0.911 | 0.855 | 0.209 | 0.712 | 0.789 |
| DIFFUSIONDB pu | DwtDct | 0.008 | 0.008 | 0.020 | 0.004 | 0.600 | 0.436 | 0.028 | 0.004 | 0.012 | 0.014 | 0.004 | 0.008 | 0.096 |
| | DwtDctSvd | 0.126 | 0.128 | 0.868 | 0.000 | 0.982 | 1.000 | 0.668 | 0.140 | 0.134 | 0.194 | 0.004 | 0.006 | 0.354 |
| | RivaGan | 0.966 | 0.972 | 0.932 | 0.012 | 0.986 | 0.990 | 0.914 | 0.050 | 0.034 | 0.106 | 0.008 | 0.010 | 0.498 |
| | SSL | 0.992 | 0.994 | 0.760 | 0.986 | 0.674 | 0.996 | 0.228 | 0.092 | 0.030 | 0.244 | 0.006 | 0.014 | 0.501 |
| | Trustmark | 1.000 | 0.996 | 0.978 | 0.000 | 0.990 | 1.000 | 0.998 | 0.892 | 0.732 | 0.020 | 0.012 | 0.006 | 0.635 |
| | VINE | 1.000 | 1.000 | 1.000 | 0.006 | 1.000 | 1.000 | 1.000 | 1.000 | 1.000 | 1.000 | 0.008 | 0.408 | 0.785 |
| | Tree-Ring | 0.682 | 0.654 | 0.546 | 0.134 | 0.534 | 0.668 | 0.610 | 0.460 | 0.498 | 0.308 | 0.048 | 0.162 | 0.474 |
| | ZoDiac | 0.936 | 0.930 | 0.874 | 0.252 | 0.844 | 0.936 | 0.922 | 0.818 | 0.794 | 0.698 | 0.088 | 0.310 | 0.702 |
| | **WF-R (Ours)** | 0.984 | 0.980 | 0.976 | 0.230 | 0.976 | 0.990 | 0.990 | 0.976 | 0.978 | 0.958 | 0.134 | 0.900 | **0.839** |
| | WF-Q (Ours) | 0.950 | 0.944 | 0.936 | 0.200 | 0.940 | 0.966 | 0.964 | 0.934 | 0.940 | 0.868 | 0.080 | 0.746 | 0.789 |
| WIKIART | DwtDct | 0.002 | 0.002 | 0.018 | 0.004 | 0.670 | 0.410 | 0.004 | 0.000 | 0.014 | 0.008 | 0.002 | 0.000 | 0.095 |
| | DwtDctSvd | 0.170 | 0.166 | 0.914 | 0.000 | 1.000 | 1.000 | 0.760 | 0.224 | 0.190 | 0.198 | 0.000 | 0.014 | 0.386 |
| | RivaGan | 0.984 | 0.982 | 0.982 | 0.002 | 0.988 | 0.990 | 0.980 | 0.074 | 0.054 | 0.182 | 0.002 | 0.004 | 0.519 |
| | SSL | 1.000 | 1.000 | 0.888 | 0.996 | 0.802 | 1.000 | 0.380 | 0.044 | 0.122 | 0.320 | 0.010 | 0.006 | 0.547 |
| | Trustmark | 0.990 | 0.984 | 0.962 | 0.012 | 0.990 | 0.998 | 0.994 | 0.846 | 0.646 | 0.026 | 0.010 | 0.002 | 0.622 |
| | VINE | 1.000 | 1.000 | 1.000 | 1.000 | 1.000 | 1.000 | 1.000 | 1.000 | 0.012 | 1.000 | 0.018 | 0.404 | 0.822 |
| | Tree-Ring | 0.610 | 0.644 | 0.470 | 0.130 | 0.492 | 0.630 | 0.514 | 0.398 | 0.398 | 0.404 | 0.054 | 0.146 | 0.435 |
| | ZoDiac | 0.906 | 0.898 | 0.852 | 0.214 | 0.834 | 0.920 | 0.828 | 0.616 | 0.678 | 0.704 | 0.084 | 0.302 | 0.653 |
| | **WF-R (Ours)** | 0.994 | 0.990 | 0.978 | 0.558 | 0.988 | 0.992 | 0.986 | 0.980 | 0.974 | 0.970 | 0.550 | 0.914 | **0.906** |
| | WF-Q (Ours) | 0.956 | 0.952 | 0.944 | 0.468 | 0.942 | 0.958 | 0.960 | 0.936 | 0.952 | 0.912 | 0.320 | 0.744 | 0.837 |

Table 5: Perceptual metrics for loss weight ablation on the DiffusionDB dataset. We highlight the best value for each metric.

| Loss Weight | PSNR ↑ | SSIM ↑ | LPIPS ↓ |
|---|---|---|---|
| $10^{-2}$ | 25.13 | **0.92** | 0.121 |
| $10^{-3}$ | 25.41 | **0.92** | 0.121 |
| $10^{-4}$ | 25.58 | **0.92** | 0.118 |
| $10^{-5}$ | **25.74** | **0.92** | **0.112** |
| $10^{-6}$ | **25.74** | **0.92** | **0.112** |

Table 6: WDR robustness metrics for loss weight ablation on the DiffusionDB dataset. We highlight the best value for each metric.

| Loss Weight | Pre-Attack ↑ | Brightness ↑ | Contrast ↑ | JPEG ↑ | Rotation ↑ | G-Noise ↑ | G-Blur ↑ | BM3D ↑ | Bmshj18 ↑ | Cheng20 ↑ | Zhao 23 ↑ | All ↑ | All + No Rotation ↑ |
|---|---|---|---|---|---|---|---|---|---|---|---|---|---|
| $10^{-2}$ | **0.991** | **0.940** | **0.950** | **0.930** | **0.580** | **0.950** | **0.980** | **0.980** | **0.910** | **0.920** | **0.900** | **0.380** | **0.710** |
| $10^{-3}$ | 0.970 | 0.850 | 0.870 | 0.810 | 0.140 | 0.830 | 0.860 | 0.830 | 0.760 | 0.780 | 0.720 | 0.090 | 0.510 |
| $10^{-4}$ | 0.957 | 0.810 | 0.810 | 0.760 | 0.110 | 0.770 | 0.840 | 0.750 | 0.710 | 0.710 | 0.620 | 0.030 | 0.350 |
| $10^{-5}$ | 0.949 | 0.780 | 0.770 | 0.700 | 0.240 | 0.680 | 0.760 | 0.740 | 0.650 | 0.610 | 0.600 | 0.140 | 0.320 |
| $10^{-6}$ | 0.940 | 0.780 | 0.770 | 0.720 | 0.230 | 0.710 | 0.770 | 0.740 | 0.630 | 0.610 | 0.600 | 0.110 | 0.320 |

Table 7: AUC results for loss weight ablation across different perturbations using DiffusionDB. We highlight the best value for each metric.

| Method | AUC (Post-Attack) | | | | | | | | | | | |
|---|---|---|---|---|---|---|---|---|---|---|---|---|
| | Brightness | Contrast | JPEG | Rotation | G-Noise | G-Blur | BM3D | Bmshj18 | Cheng20 | Zhao23 | All | All w/o Rot. |
| $10^{-2}$ | 0.991 | 0.997 | 0.993 | 0.881 | 0.994 | 0.997 | 0.996 | 0.994 | 0.985 | 0.987 | 0.806 | 0.947 |
| $10^{-3}$ | 0.984 | 0.989 | 0.980 | 0.731 | 0.982 | 0.986 | 0.987 | 0.977 | 0.953 | 0.961 | 0.636 | 0.904 |
| $10^{-4}$ | 0.975 | 0.980 | 0.976 | 0.714 | 0.966 | 0.979 | 0.976 | 0.956 | 0.938 | 0.944 | 0.576 | 0.831 |
| $10^{-5}$ | 0.951 | 0.944 | 0.923 | 0.687 | 0.902 | 0.940 | 0.916 | 0.898 | 0.875 | 0.874 | 0.559 | 0.705 |
| $10^{-6}$ | 0.951 | 0.944 | 0.928 | 0.685 | 0.890 | 0.940 | 0.918 | 0.898 | 0.875 | 0.874 | 0.564 | 0.717 |

Table 8: TPR@1%FPR results for loss weight ablation across different perturbations using DiffusionDB. We highlight the best value for each metric.

| Method | TPR@1%FPR (Post-Attack) | | | | | | | | | | | |
|---|---|---|---|---|---|---|---|---|---|---|---|---|
| | Brightness | Contrast | JPEG | Rotation | G-Noise | G-Blur | BM3D | Bmshj18 | Cheng20 | Zhao23 | All | All w/o Rot. |
| $10^{-2}$ | **0.950** | **0.950** | **0.920** | **0.470** | **0.920** | **0.940** | **0.910** | **0.910** | **0.830** | **0.830** | **0.220** | **0.800** |
| $10^{-3}$ | 0.920 | 0.890 | 0.860 | 0.210 | 0.860 | 0.860 | 0.790 | 0.810 | 0.730 | 0.730 | 0.090 | 0.590 |
| $10^{-4}$ | 0.890 | 0.880 | 0.790 | 0.170 | 0.720 | 0.800 | 0.740 | 0.780 | 0.670 | 0.640 | 0.030 | 0.310 |
| $10^{-5}$ | 0.740 | 0.680 | 0.510 | 0.150 | 0.520 | 0.540 | 0.500 | 0.570 | 0.420 | 0.500 | 0.050 | 0.220 |
| $10^{-6}$ | 0.740 | 0.670 | 0.500 | 0.150 | 0.500 | 0.560 | 0.510 | 0.550 | 0.400 | 0.500 | 0.050 | 0.160 |

## C.3 WATERMARK RADIUS ABLATION

In this ablation we modify the radius of our learned watermark and observe the corresponding results. We present our results in Table 9, 10, 11.

We observe that increasing the watermark radius leads to a more robust watermark. This makes sense as the watermark assumes more "area". However, what is slightly surprising is that the LPIPS seems to become better with a larger watermark radius. So while PSNR suffers, we can understand this as our model creating more realistic images that differ from the original image. A potential reason for this is that a larger watermark means that we have more control over the latent.

## C.4 MODEL ARCHITECTURE ABLATION

In this ablation we try various generative model architectures for parameterizing our learned watermark. We present our results in Table 12 and 13. We use the hyperparameters listed in Appendix B. We note that for these experiments we use a SSIM threshold of 0.92.

We observe that the Residual Flow architecture yields the best results in terms of robustness although UNet and MLP do slightly better on perceptual metrics. The biggest problem with the UNet architecture is that the learned watermark is simply too weak. That is, the watermarked images are not statistically separable from the non-watermarked images. This can be observed by the 50% AUC and 0 WDR. While MLP is slightly better, it still falls short of the Residual Flow Architecture.

Table 9: Perceptual and WDR metric for watermark radius ablation. We use the DiffusionDB dataset for this experiment. We highlight the best value for each metric.

| Metric | 5 | 10 | 15 | 20 |
|---|---|---|---|---|
| PSNR ↑ | **25.69** | 25.13 | 25.22 | 25.06 |
| SSIM ↑ | **0.92** | **0.92** | **0.92** | **0.92** |
| LPIPS ↓ | 0.121 | 0.121 | 0.117 | **0.095** |
| Pre-Attack ↑ | 0.958 | 0.991 | 0.998 | **0.999** |
| Brightness ↑ | 0.770 | 0.940 | **0.990** | **0.990** |
| Contrast ↑ | 0.810 | 0.950 | 0.980 | **0.990** |
| JPEG ↑ | 0.740 | 0.930 | 0.990 | **1.000** |
| Rotation ↑ | 0.290 | 0.580 | 0.340 | **0.770** |
| G-Noise ↑ | 0.810 | 0.950 | 0.980 | **0.990** |
| G-Blur ↑ | 0.850 | 0.980 | **1.000** | 0.990 |
| BM3D ↑ | 0.820 | 0.980 | **1.000** | **1.000** |
| Bmshj18 ↑ | 0.710 | 0.910 | 0.970 | **0.990** |
| Cheng20 ↑ | 0.760 | 0.920 | **0.990** | **0.990** |
| Zhao23 ↑ | 0.720 | 0.900 | 0.920 | **0.990** |
| All ↑ | 0.140 | 0.380 | 0.220 | **0.540** |
| All + No Rotation ↑ | 0.480 | 0.710 | 0.810 | **0.890** |

Table 10: AUC results for watermark radius ablation across different perturbations using DiffusionDB. We highlight the best value for each metric.

| Radius | Brightness | Contrast | JPEG | Rotation | G-Noise | G-Blur | BM3D | Bmshj18 | Cheng20 | Zhao23 | All | All w/o Rot. |
|---|---|---|---|---|---|---|---|---|---|---|---|---|
| 5 | 0.977 | 0.980 | 0.970 | 0.798 | 0.975 | 0.974 | 0.980 | 0.957 | 0.964 | 0.949 | 0.712 | 0.885 |
| 10 | 0.991 | 0.997 | 0.993 | 0.881 | 0.994 | 0.997 | 0.996 | 0.994 | 0.985 | 0.987 | 0.806 | 0.947 |
| 15 | **1.000** | **1.000** | 0.999 | 0.850 | **0.998** | **1.000** | **1.000** | **1.000** | **0.999** | 0.994 | 0.806 | **0.969** |
| 20 | 0.999 | **1.000** | **1.000** | **0.960** | 0.996 | **1.000** | **1.000** | 0.999 | **0.999** | **0.996** | **0.874** | 0.963 |

Table 11: TPR@1%FPR results for watermark radius ablation across different perturbations using DiffusionDB. We highlight the best value for each metric.

| Radius | Brightness | Contrast | JPEG | Rotation | G-Noise | G-Blur | BM3D | Bmshj18 | Cheng20 | Zhao23 | All | All w/o Rot. |
|---|---|---|---|---|---|---|---|---|---|---|---|---|
| 5 | 0.680 | 0.770 | 0.640 | 0.280 | 0.670 | 0.780 | 0.810 | 0.690 | 0.490 | 0.570 | 0.220 | 0.370 |
| 10 | 0.950 | 0.950 | 0.920 | 0.470 | 0.920 | 0.940 | 0.910 | 0.910 | 0.830 | 0.830 | 0.220 | 0.800 |
| 15 | **1.000** | **1.000** | **0.990** | **0.500** | 0.980 | **1.000** | **1.000** | **0.990** | **0.990** | 0.970 | 0.300 | **0.850** |
| 20 | 0.990 | 0.990 | **0.990** | 0.360 | **0.990** | **1.000** | **1.000** | **0.990** | **0.990** | **0.990** | **0.450** | **0.850** |

## C.5 A NOTE ON INITIALIZING PATCH TO TREE-RING

We found in earlier iterations of our work that not initializing with a tree-ring patch produced significantly worse results. Furthermore, adding the tree-ring patch to the learned patch was also worse. This is founded on the hypothesis that the latent with the tree-ring watermark is already a strong starting point and our mapping simply adjusts it as needed to trade off robustness and quality.

## C.6 VARYING SSIM THRESHOLD

Our results are shown in Figure 7. We present an additional ablation which involves varying the SSIM threshold used for adaptive enhancement. We obviously expect the image quality metrics to get better as we increase the SSIM threshold (note both the PSNR and LPIPS metric). The perhaps more surprising story is the little drop-off in robustness. Between an SSIM of 0.92 and 0.95, there is little to no drop in robustness quality. The difference starts to become slightly more noticeable when we go up to 0.99 but is still relatively good compare to our baselines. We hypothesize that because adaptive enhancement is similar to an adversarial attack (though beneficial for us in this case), it does not deter WaterFlow which is by nature incredibly robust.

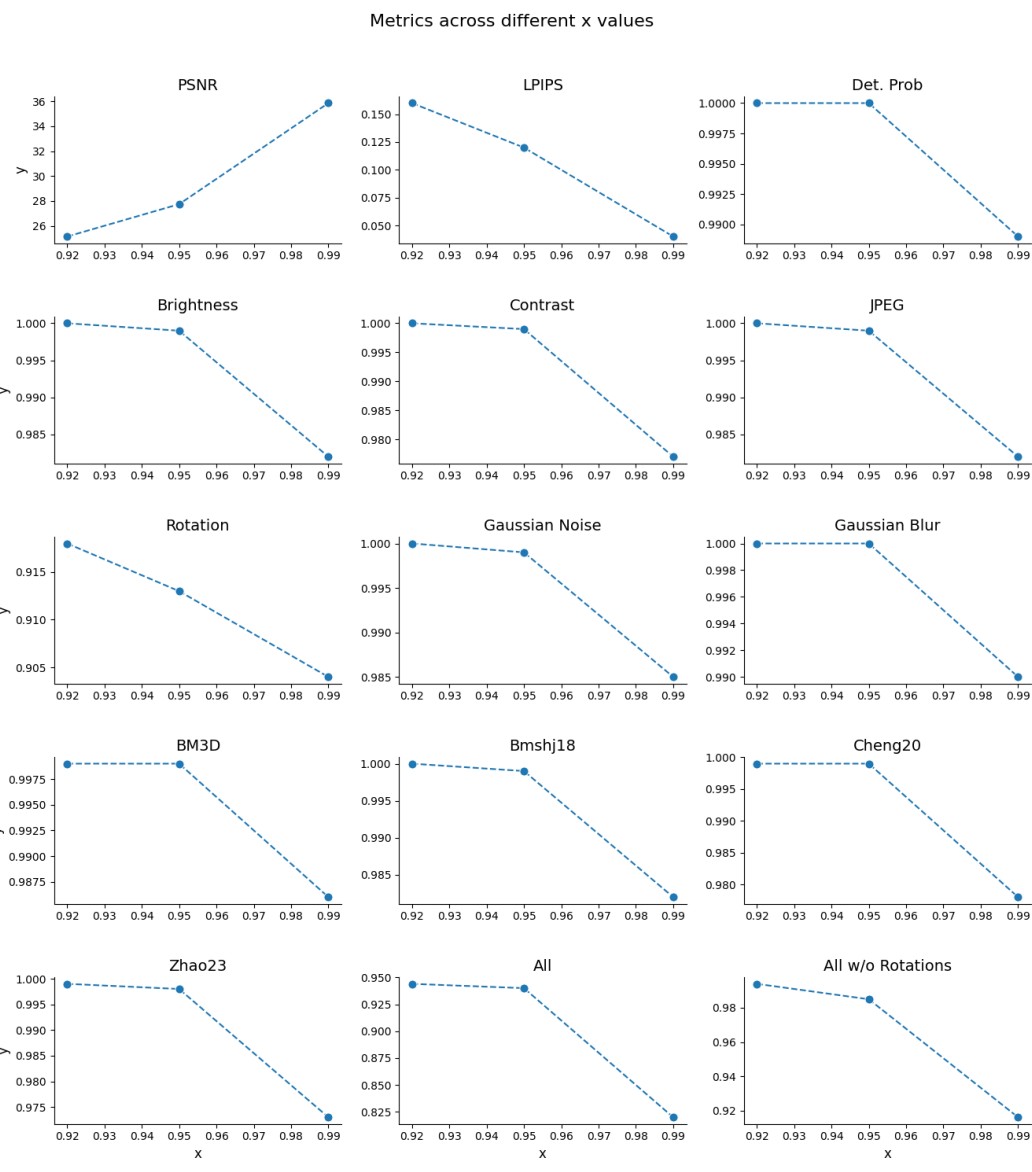

Figure 7: Results showing image quality and robustness as a function of various SSIM thresholds used for adaptive enhancement. In this figure, all robustness metrics are in AUC and the thresholds we test are $0.92, 0.95, 0.99$.

Table 12: Perceptual and WDR metrics for model architecture ablation (DiffusionDB). Rows are metrics, columns are model architectures. We highlight the best value for each metric.

| Metric | MLP | Residual Flow | UNet |
|---|---|---|---|
| PSNR ↑ | 25.91 | 25.13 | **26.00** |
| SSIM ↑ | **0.92** | **0.92** | **0.92** |
| LPIPS ↓ | 0.111 | 0.121 | **0.107** |
| Pre-Attack ↑ | 0.908 | **0.991** | 0.501 |
| Brightness ↑ | 0.630 | **0.940** | 0 |
| Contrast ↑ | 0.600 | **0.950** | 0 |
| JPEG ↑ | 0.410 | **0.930** | 0 |
| Rotation ↑ | 0.080 | **0.580** | 0 |
| G-Noise ↑ | 0.520 | **0.950** | 0 |
| G-Blur ↑ | 0.630 | **0.980** | 0 |
| BM3D ↑ | 0.540 | **0.980** | 0 |
| Bmshj18 ↑ | 0.460 | **0.910** | 0 |
| Cheng20 ↑ | 0.420 | **0.920** | 0 |
| Zhao23 ↑ | 0.430 | **0.900** | 0 |
| All ↑ | 0.000 | **0.380** | 0 |
| All + No Rotation ↑ | 0.090 | **0.710** | 0 |

Table 13: Results for model architecture ablation. We show AUC and TPR@1%FPR across a wide variety of different attacks and perturbations. The dataset used is DiffusionDB. We highlight the best value for each metric.

| Method | Post-Attack | | | | | | | | | | | |
|---|---|---|---|---|---|---|---|---|---|---|---|---|
| | Brightness | Contrast | JPEG | Rotation | G-Noise | G-Blur | BM3D | Bmshj18 | Cheng20 | Zhao23 | All | All w/o Rot. |
| MLP | 0.922/0.554 | 0.910/0.485 | 0.863/0.386 | 0.540/0.030 | 0.880/0.505 | 0.919/0.495 | 0.900/0.465 | 0.841/0.505 | 0.827/0.396 | 0.854/0.327 | 0.485/0.000 | 0.663/0.069 |
| Residual Flow | **0.991/0.950** | **0.997/0.950** | **0.993/0.920** | **0.881/0.470** | **0.994/0.920** | **0.997/0.940** | **0.996/0.910** | **0.994/0.910** | **0.985/0.830** | **0.987/0.830** | **0.806/0.220** | **0.947/0.800** |
| UNet | 0.553/0.030 | 0.538/0.050 | 0.541/0.000 | 0.499/0.020 | 0.539/0.020 | 0.562/0.010 | 0.518/0.000 | 0.493/0.000 | 0.514/0.000 | 0.550/0.000 | 0.494/0.010 | 0.536/0.000 |

Table 14: Perceptual similarity (PSNR ↑, LPIPS ↓) and watermark detection under all attacks AUC (Full Avg. ↑) for WaterFlow-General. Best values per metric are **bold**.

| Dataset | Method | PSNR ↑ | LPIPS ↓ | AUC ↑ |
|---|---|---|---|---|
| **COCO** | WaterFlow-General | 27.82 | **0.07** | 0.945 |
| | ZoDiac | **28.61** | 0.13 | 0.937 |
| | WaterFlow | 27.74 | 0.12 | **0.984** |
| **DiffDB** | WaterFlow-General | 27.69 | **0.07** | 0.946 |
| | ZoDiac | **28.65** | 0.11 | 0.937 |
| | WaterFlow | 27.39 | 0.09 | **0.985** |
| **WikiArt** | WaterFlow-General | 28.04 | **0.09** | 0.945 |
| | ZoDiac | **28.93** | 0.10 | 0.923 |
| | WaterFlow | 26.94 | 0.10 | **0.982** |

## C.7 GENERALIZED WATERFLOW

We conduct an experiment where WaterFlow is trained jointly on all three datasets, using 100 samples from each. Our results are shown in Table 15 and Table 16. We then separately evaluate performance on each dataset. Our results show that WaterFlow-General achieves higher perceptual quality than the original WaterFlow and performs comparably to ZoDiac. Additionally, while WaterFlow-General surpasses ZoDiac in robustness, it remains slightly less robust than the original WaterFlow. This shows that even in the case where we must deploy a single mode to be used on a wide variety of images, WaterFlow remains a strong method. We note that given the expansive nature of our datasets as well as how lightweight our models are, it may make sense to actually train multiple models to learn domain-specific watermarks which promise much higher robustness.

Table 15: Perceptual similarity metrics (SSIM ↑, PSNR ↑, LPIPS ↓) across three datasets for generalizability experiments. We bold the best values in each dataset.

| Dataset | Method | SSIM ↑ | PSNR ↑ | LPIPS ↓ |
|---|---|---|---|---|
| **COCO** | WaterFlow-General | **0.95** | 27.82 | **0.07** |
| | ZoDiac | 0.92 | **28.61** | 0.13 |
| | WaterFlow | **0.95** | 27.74 | 0.12 |
| **DiffDB** | WaterFlow-General | **0.95** | 27.69 | **0.07** |
| | ZoDiac | 0.92 | **28.65** | 0.11 |
| | WaterFlow | **0.95** | 27.39 | 0.09 |
| **WikiArt** | WaterFlow-General | **0.95** | 28.04 | **0.09** |
| | ZoDiac | 0.92 | **28.93** | 0.10 |
| | WaterFlow | **0.95** | 26.94 | 0.10 |

Table 16: AUC of watermark detection after various image perturbations generalizability experiment. We bold the best full average AUC per dataset.

| Dataset | Method | Brightness | Contrast | JPEG | Rotation | G-Noise | G-Blur | BM3D | Bmshj18 | Cheng20 | Zhao23 | All | w/o Rot. | Full Avg. |
|---|---|---|---|---|---|---|---|---|---|---|---|---|---|---|
| COCO | ZoDiac | 0.994 | 0.994 | 0.989 | 0.800 | 0.989 | 0.996 | 0.991 | 0.988 | 0.984 | 0.960 | 0.622 | 0.836 | 0.937 |
| | WaterFlow | 1.000 | 1.000 | 1.000 | 0.913 | 0.999 | 1.000 | 0.999 | 0.999 | 0.999 | 0.998 | 0.879 | 0.967 | **0.984** |
| | WaterFlow-General | 0.988 | 0.988 | 0.987 | 0.852 | 0.980 | 0.983 | 0.974 | 0.966 | 0.971 | 0.972 | 0.778 | 0.905 | 0.945 |
| DiffDB | ZoDiac | 0.994 | 0.994 | 0.989 | 0.800 | 0.989 | 0.996 | 0.991 | 0.988 | 0.984 | 0.960 | 0.622 | 0.836 | 0.937 |
| | WaterFlow | 1.000 | 1.000 | 1.000 | 0.903 | 0.999 | 1.000 | 1.000 | 1.000 | 0.999 | 0.999 | 0.879 | 0.967 | **0.985** |
| | WaterFlow-General | 0.988 | 0.993 | 0.985 | 0.851 | 0.983 | 0.993 | 0.993 | 0.980 | 0.982 | 0.981 | 0.787 | 0.920 | 0.946 |
| WikiArt | ZoDiac | 0.991 | 0.991 | 0.981 | 0.732 | 0.988 | 0.993 | 0.984 | 0.964 | 0.960 | 0.956 | 0.572 | 0.812 | 0.923 |
| | WaterFlow | 0.997 | 0.998 | 0.999 | 0.937 | 1.000 | 0.999 | 0.999 | 0.996 | 0.996 | 0.995 | 0.896 | 0.986 | **0.982** |
| | WaterFlow-General | 0.992 | 0.992 | 0.990 | 0.868 | 0.985 | 0.988 | 0.983 | 0.973 | 0.977 | 0.978 | 0.796 | 0.941 | 0.945 |

## C.8 ADVERSARIAL EMBEDDING ATTACKS

We present results in Table 17. While the WAVES paper introduces a broad set of attacks, our evaluation covers a similarly diverse range, including additional combination attacks that merge multiple perturbation types. One category we did not initially include is adversarial embedding attacks, which specifically target the model's embedding space. Although these attacks are theoretically promising, we observe that even traditionally weak watermarking methods perform unexpectedly well against them, casting doubt on their practical effectiveness. Thus, we observe that most methods achieve very high AUC. We believe that this is because they are ill-suited as an attack when the image quality is sufficiently good. Nonetheless, we include comparative results for completeness.

## D  LIMITATIONS

Our method builds on pre-trained Stable Diffusion models, which, while powerful across many domains, may not generalize well to domain-specific imagery such as medical scans or satellite data. Since our approach assumes access to a known latent space and invertible generative process, it may not be directly applicable to other state-of-the-art models that are non-invertible or autoregressive in nature. Furthermore, we focus on a single open-source model, and the transferability of our method to closed-source or fine-tuned proprietary diffusion models remains uncertain. Finally, we observe that in some cases, the watermark introduces slight visual artifacts. However, this trade-off can be controlled via loss balancing and adaptive enhancement techniques.

## E  BROADER IMPACT

As generative models continue to be adopted for content creation, the need for robust watermarking mechanisms becomes increasingly urgent—both to ensure accountability and to mitigate the spread of synthetic misinformation. Our method, WaterFlow, offers a promising step toward reliable, high-fidelity watermarking of AI-generated images. It is lightweight, robust to a wide range of perturbations, and fast enough for practical deployment.

Table 17: AUC for different adversarial embedding attacks. Values greater than 0.95 are highlighted in bold to represent strong resistance against the attack.

| Method | AdvEmbG-KLVAE8 | AdvEmbB-CLIP | ResNet18 | AdvEmbB-SdxlVAE |
|---|---|---|---|---|
| DwtDct | 0.948 | **0.952** | 0.948 | 0.944 |
| DwtDctSvd | **0.991** | **0.961** | **0.980** | **0.994** |
| RivaGAN | **1.000** | **1.000** | **0.997** | **1.000** |
| SSL | 0.947 | **0.972** | **0.963** | 0.875 |
| TrustMark | **0.966** | 0.860 | 0.867 | **0.990** |
| VINE | **1.000** | **1.000** | **1.000** | **1.000** |
| ZoDiac | 0.677 | **0.962** | **0.958** | **0.954** |
| WF-R | **0.970** | **0.995** | **0.993** | **0.995** |
| WF-Q | 0.929 | **0.978** | **0.979** | **0.984** |
| DwtDct | 0.912 | 0.922 | 0.918 | 0.903 |
| DwtDctSvd | **0.978** | 0.945 | **0.960** | **0.981** |
| RivaGAN | **0.999** | **0.999** | **0.999** | **1.000** |
| SSL | 0.895 | **0.957** | 0.927 | 0.828 |
| TrustMark | **0.973** | 0.868 | 0.883 | **0.993** |
| VINE | **1.000** | **1.000** | **1.000** | **1.000** |
| ZoDiac | 0.741 | **0.963** | **0.960** | **0.958** |
| WF-R | **0.978** | **0.996** | **0.992** | **0.994** |
| WF-Q | 0.949 | **0.985** | **0.985** | **0.987** |
| DwtDct | 0.919 | 0.924 | 0.920 | 0.916 |
| DwtDctSvd | **0.967** | **0.981** | **0.986** | **0.997** |
| RivaGAN | **0.997** | **0.996** | **0.998** | **0.998** |
| SSL | 0.907 | **0.973** | **0.963** | 0.875 |
| TrustMark | **0.963** | 0.867 | 0.920 | **0.994** |
| VINE | **1.000** | **1.000** | **1.000** | **1.000** |
| ZoDiac | 0.618 | **0.970** | **0.956** | **0.963** |
| WF-R | **0.971** | **0.999** | **0.998** | **0.998** |
| WF-Q | 0.925 | **0.991** | **0.990** | **0.989** |

However, any watermarking system also introduces ethical considerations. Malicious actors may adapt these techniques to embed imperceptible information in ways that violate user privacy or evade detection. Conversely, watermark removal techniques may evolve in tandem, creating an ongoing arms race. It is crucial that watermarking research is accompanied by transparent reporting, open benchmarking, and active dialogue between academia, industry, and policymakers. We hope this work contributes positively to the development of ethical and trustworthy generative AI systems.

## F LLM USAGE

LLMs were used to help with manuscript writing.

## G REPRODUCIBILITY

We include a list of hyperparameters, hardware, and other training configurations in the appendix. Furthermore, we include the source code as part of the supplementary material.

## H FUTURE WORK

Future directions include expanding evaluation to a wider range of datasets and real-world attack scenarios to better understand resilience under adversarial or noisy conditions. While our current design assumes the generative model's latent space is fixed, future research could explore fine-tuning diffusion backbones jointly with the watermarking objective. We are also interested in adapting WaterFlow to other generative paradigms, including autoregressive or transformer-based models, and in developing defenses against emerging watermark-removal attacks. Finally, we plan to explore learnable post-processing steps that further improve fidelity while preserving robustness.

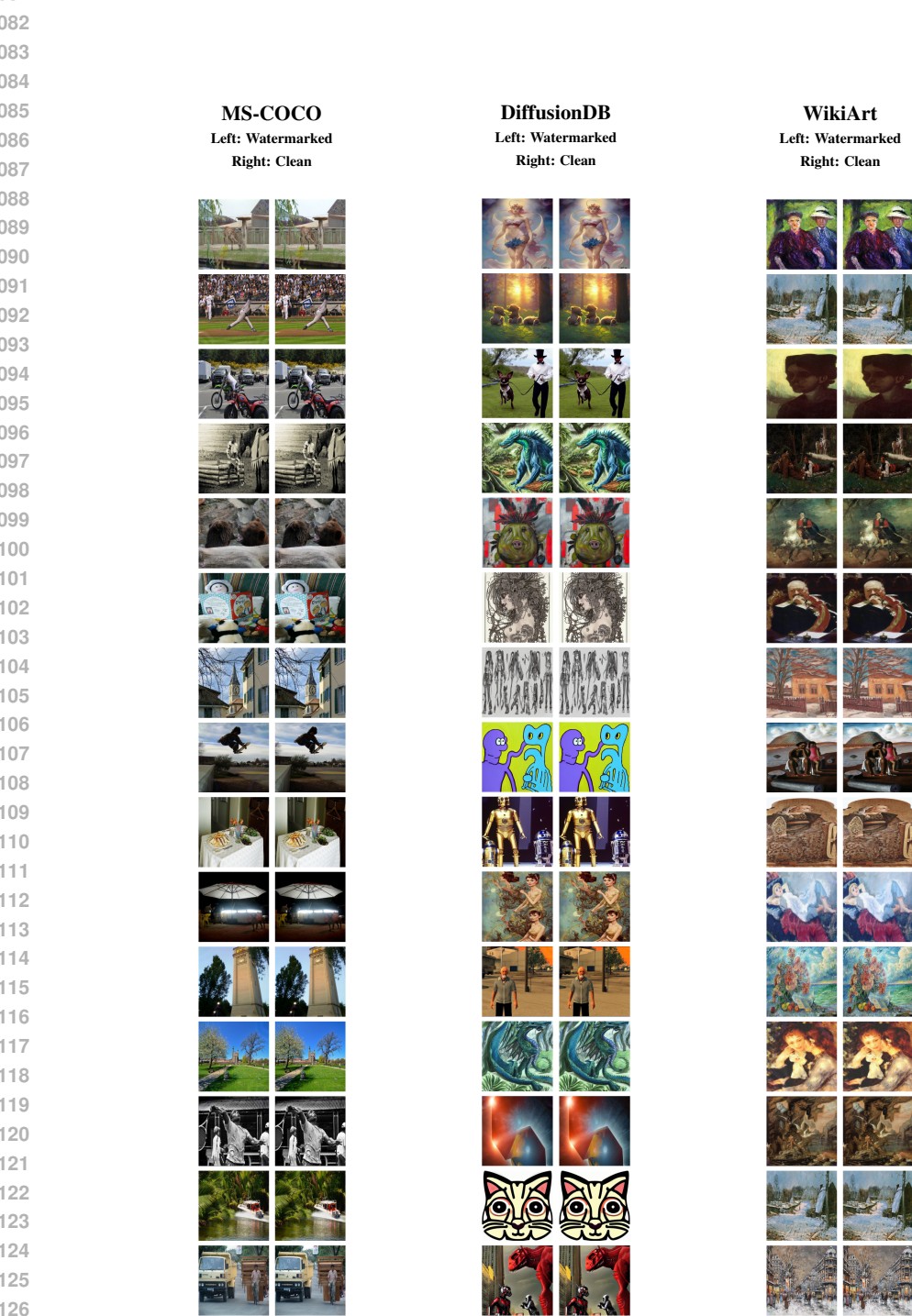

Figure 8: Example images from MS-COCO, DiffusionDB, and WikiArt datasets. Each pair shows a watermarked (left) and clean (right) image.

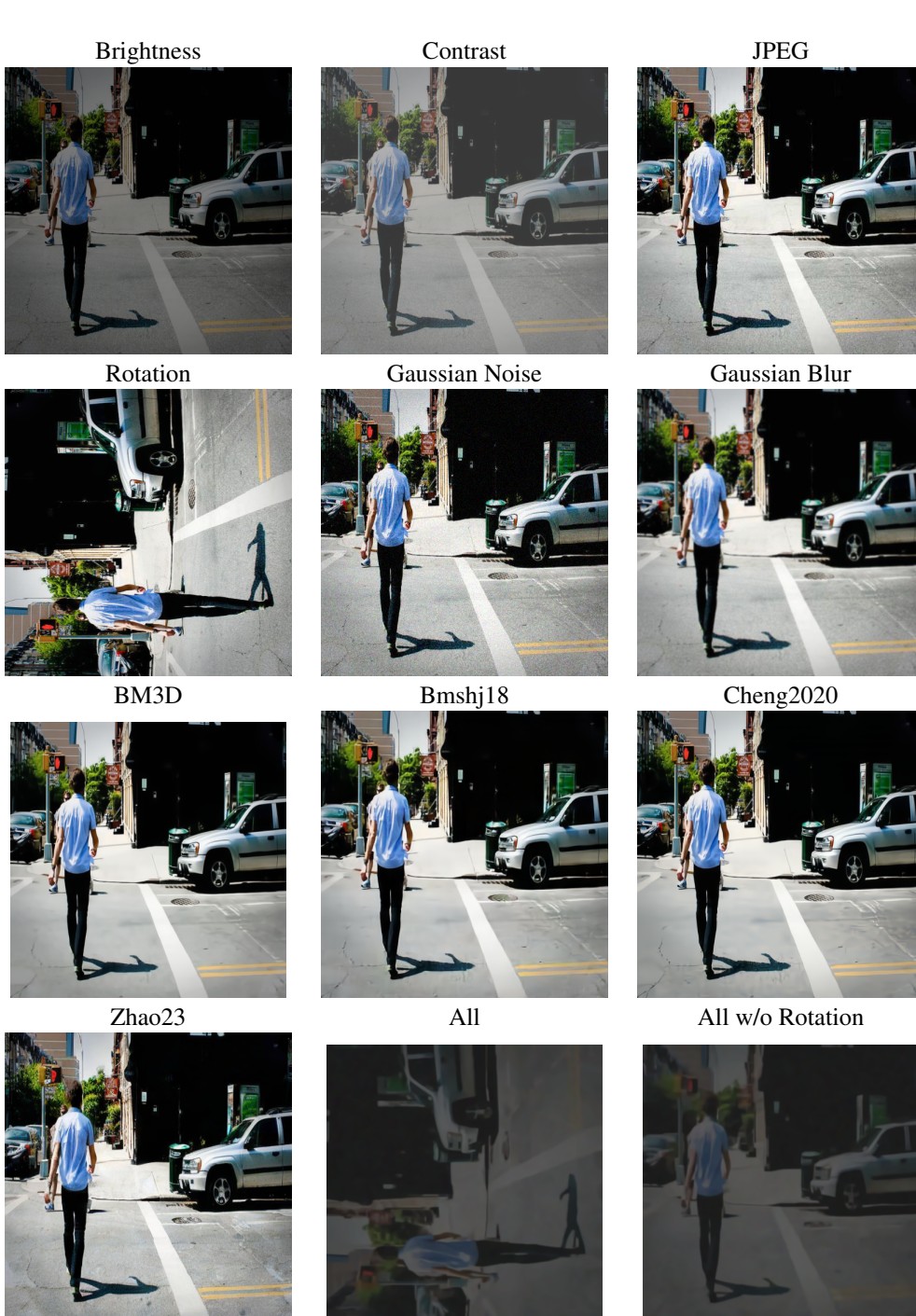

Figure 9: Examples of attacks on watermarked image from MS-COCO dataset.

