# OpenReview forum: "WaterFlow: Fast and Robust Watermarking in the Latent Fourier Domain"
_ICLR.cc/2026/Conference — Submitted to ICLR 2026_

### Official Review · Reviewer_W4xZ · 2025-10-15

**Soundness:** 2
**Presentation:** 3
**Contribution:** 1
**Rating:** 2
**Confidence:** 5

**Summary:**

This paper introduces WaterFlow (WF), a latent watermarking approach that, learns a trainable watermark per-image in the Fourier domain. The key contribution lies in Section 3.2, where the authors propose directly optimizing the watermark signal using a new dissimilarity loss, which encourages the watermark to differ from the underlying latent content. The paper claims improvements in both speed and robustness, especially under compound or combined attacks, while maintaining visual quality.

**Strengths:**

The paper proposes a simple yet practical shift in watermarking strategy—from optimizing the latent representation to directly learning a trainable watermark. This change has the potential to improve watermark robustness.

**Weaknesses:**

The central issue with this paper lies in questionable novelty and insufficient motivation. As far as I can tell, the only truly novel component is the idea in Section 3.2—to optimize the watermark itself, rather than optimizing the watermarked latent vector (as ZoDiac does). While this idea is potentially useful, the paper fails to convincingly articulate why this is a meaningful or necessary innovation, or under what conditions it would yield substantial benefits. This makes it difficult to assess whether the work meets the contribution threshold expected for a top-tier conference.

In particular, I found the motivation in the Introduction unconvincing. The authors state that their method “learns a watermark in the Fourier domain of the latent representation—enhancing robustness while preserving perceptual quality,” and that it “supports control over the quality–robustness trade-off without retraining.” However, both of these claims already apply to ZoDiac. What, then, is new here? The Introduction also claims that this is “the first watermarking method to withstand complex combination attacks,” but the method proposed in Section 3.2 is not clearly connected to this claim. There is a clear disconnect between what the method actually does (optimize W* with an additional dissimilarity loss) and what it claims to achieve (faster, more robust watermarking under compound attacks). A stronger, evidence-backed explanation of this link is sorely needed.

**Questions:**

### Major Concerns

1. Why is Tree-Ring still used in the optimized watermark?
One of the key claims of this paper is that it learns a trainable watermark W*. If the method is designed to learn the optimal watermark, why is it still constrained to include a Tree-Ring pattern?

2. How about using gradient-based optimization on W* directly?
Section 5.2 explores different neural networks to generate W*. However, if W* is image-specific and trainable, why not simply optimize it per image via backpropagation, as in ZoDiac, but with the additional dissimilarity loss? This would test whether the benefit comes from the new objective or the model-based watermark generator.

3. The runtime comparison seems misleading.
My most critical concern is the claim that WF is significantly faster than ZoDiac due to eliminating per-image latent optimization. Figure 3 claims that WF achieves faster watermarking since it applies a single universal forward pass, unlike ZoDiac’s iterative optimization. However, Section 3.2 clearly states that the trainable watermark W*  is adapted per image, and is generated based on a loss function, suggesting that some form of per-image optimization still occurs. If that is the case, why is the latency comparison not accounting for this per-image cost? The fairness and correctness of this comparison need to be clarified. Otherwise, the claimed speed advantage may be overstated or misleading.

### Minor Issues
1. Figure 1 lacks context.
What kind of attack is used to generate the robustness AUC shown in Figure 1? This figure appears prominently in the introduction but lacks a clear experimental setting.

2. Why use a circular binary mask in Eq. (8)?
Similar to ZoDiac and Tree-Ring, WF uses a circular binary mask to modulate W*. If WF is claimed to learn a flexible and trainable watermark, why constrain it with a hand-designed masking prior? Is this necessary for robustness, or is it simply inherited from prior work?

---

### Official Review · Reviewer_EkDk · 2025-10-24

**Soundness:** 3
**Presentation:** 2
**Contribution:** 3
**Rating:** 4
**Confidence:** 4

**Summary:**

This paper introduces WaterFlow, a watermarking framework for images that operates in the latent Fourier domain of pretrained diffusion models. It embeds adaptive watermarks using a lightweight flow-based generator trained to balance imperceptibility and robustness, initializing with a Tree-Ring pattern and injecting into the latent frequency space. Two variants, WF-R (focus on robustness) and WF-Q (focus on quality), allow trade-offs via a postprocessing parameter without retraining. Evaluations demonstrate superior robustness under various attacks compared to baselines, while maintaining competitive image quality.

**Strengths:**

1. Good Engineering Design: The research demonstrates good originality by combining frequency-domain watermarking in latent spaces with a trainable, image-adaptive watermark generator that utilizes residual flows, introducing a novel separability loss term to enhance detectability without requiring per-image optimization. This creative design enables faster embedding, suitable for real-world applications.

2. Comprehensive Robustness Validation: The authors provide strong support for the method's practical effectiveness by conducting extensive experiments and a thorough robustness analysis. This analysis is comprehensive, evaluating the method against traditional perturbations, advanced diffusion-based generative attacks, and, most notably, aggressive compound attacks. As the results demonstrate, the paper's most significant strength is its unparalleled robustness against these composite attacks.

**Weaknesses:**

1. Poor Paper Writing: Some sections of the methodology are difficult to follow, particularly for non-experts in the field. Eq. 8 lacks an explanation, which may leave readers questioning. The authors state that they "do not guarantee Hermitian symmetry anymore," yet they do not explain what this means to FFT or what it is before making the claim. Additionally, the symbols used should be defined within the context of the paper. For instance, I would not know what $\Re$ and $\Im$ refer to in Eq. 7 unless I reread the previous words. The authors are using the concepts without introducing or defining them first.

2. Unclear Contribution Relative to Tree-Ring Initialization: The method's dependency on an initial Tree-Ring watermark ($W_{Tree}$) is a notable weakness. The appendix explicitly states that not using this initialization "produced significantly worse results". This suggests that WaterFlow is perhaps best described as a refinement or enhancement of Tree-Ring, rather than a completely independent watermarking scheme.

**Questions:**

Can you explain why you selected these losses? What reason or purpose does each loss stand for? And why do you use the comprehensive loss design?

Why is the learned watermark only applied to the last channel? What is special about this channel?

Can the authors provide more quantitative results about their claim, “This is founded on the hypothesis that the latent with the tree-ring watermark is already a strong starting point and our mapping simply adjusts it as needed to trade off robustness and quality.”

---

### Official Review · Reviewer_ASCj · 2025-10-31

**Soundness:** 3
**Presentation:** 3
**Contribution:** 2
**Rating:** 4
**Confidence:** 4

**Summary:**

This paper proposes WaterFlow, a latent-space watermarking framework that embeds signals in the Fourier domain of diffusion model latents. Unlike existing approaches that rely on end-to-end training or per-image optimization, WaterFlow trains a lightweight flow-based generator once, and then performs fast, adaptive watermark injection into the latent frequency domain. The method claims to provide strong robustness against generative removal attacks and complex compound distortions, while maintaining high perceptual fidelity and offering a tunable quality-robustness trade-off without retraining. Experiments on MS-COCO, DiffusionDB, and WikiArt show competitive image quality and improved robustness over prior works such as Tree-Ring, ZoDiac, and VINE.

**Strengths:**

1. Working directly in the latent frequency domain with diffusion-space inversion is well-motivated and elegant. The combination of Tree-Ring initialization + learned Fourier-space perturbation is original and technically interesting.

2. Practical and efficient. No per-image optimization. Adjustable quality-robustness trade-off without retraining is a useful feature for deployment.

3. Strong robustness against regeneration and compound perturbations. The method consistently outperforms classical and diffusion watermarking methods under composite noise + generative attacks.

**Weaknesses:**

1. Limited novelty in watermark embedding strategy.
The core idea—injecting watermarks in the latent Fourier domain—is not new and has been explored in prior works such as Tree-Ring and ZoDiac. This method mainly refines and engineers existing concepts rather than providing a fundamentally novel watermarking mechanism.

2. Incomplete baseline comparison with recent stronger methods.
The paper does not benchmark against several state-of-the-art watermarking approaches, including Watermark Adapter and Gaussian Shading watermarking, which limits the strength of empirical claims. Comparisons are mostly against older baselines, making it difficult to judge true progress.

3. Quality-robustness trade-off is not genuinely continuous or unified.
Although the paper claims controllability between watermark quality and robustness, the two variants (WF-R and WF-Q) behave like separate configurations rather than a smooth, unified control mechanism. This indicates that the method struggles to simultaneously maintain high fidelity and strong robustness in a single model, reducing practical flexibility.

**Questions:**

How robust is your method against more advanced reconstruction or removal attacks (e.g., ControlGen 【1】 or other conditional re-generation models )? Could you provide quantitative comparisons against such attacks

1. Image Watermarks are Removable Using Controllable Regeneration from Clean Noise

---

### Official Review · Reviewer_MrTA · 2025-11-04

**Soundness:** 3
**Presentation:** 3
**Contribution:** 3
**Rating:** 4
**Confidence:** 4

**Summary:**

The paper introduces a new invisible watermarking approach for images. The idea is to use the pre-trained latent diffusion models and learn the watermark model in the Fourier domain of the latent representation. The claim is that the resulting model is light-weight (10^4 learnable parameters) and enhancing robustness while preserving visual quality. The paper is evaluated using three different datasets and compared against eight different baselines and ten different synthetic attack types.

**Strengths:**

- The idea of using FFT on the latent space is considered novel and technically sound
- The paper is well-written, including extensive experiments/ablations.
- Discussing multiple recent baselines and evaluations across three different datasets.

**Weaknesses:**

1- Frequency based watermarking approaches are known to be sensitive against geometric transformations such as resize and crop (in addition to rotations, overlay text/emoji etc). It is not clear whether these weaknesses exist for the proposed method or not. Additional ablations and attack vectors necessary. It is advised to focus on real-world attack vectors such as resize, crop, bordering, text/emoji overlay, horizontal flip and their combinations.
2- The visual quality metrics SSIM / PSNR are not as high as the baseline methods. Therefore the claim that "visual quality is preserved" does not have an empirical support
3- G-Blur attacks (5x5 kernel with std = 1) is considerably weak. All known watermarking methods are expected provide very high accuracy (as also shown in Table 1). No need to include this attack vector. Rotation by 90 degrees is also not a relevant/common real-world attack vector.
4 - The efficiency evaluations lacks necessary details for repeatability and interpretability. WaterFlow's 6s latency needs to be contextualized. Otherwise the efficiency claims remain weak. Some of the baseline methods have similar efficiency numbers, therefore not clear if the proposed solution has a particular advantage over other baselines. Commercial watermarking methods provide <50ms latency on CPUs for 4MP imagery.
5- It is impossible to evaluate the visual quality of the watermarked images based on the thumbnail images in the appendix. The links to the original images are required for a subject visual quality evaluation.

**Questions:**

- Image and video invisible watermarking problems are dual problems. Would the proposed approach generalize for video watermarking problem?  What would be the pros and cons of using the latent diffusion models compared to other methods such as VideoSeal? Any discussions on how to address both problems using the proposed method?
- The  advantages of the proposed method along the visual quality and efficiency axes are not strongly justified/proven experimentally. The attack vectors are also somewhat weak, therefore I would like to hear from the authors and other reviewers before up-leveling the paper's rating.

---

### Meta-Review · Area_Chair_nLzQ · 2026-01-05

**Summary:**

The paper proposes WaterFlow, a latent-space image watermarking method for diffusion models that embeds watermarks in the Fourier domain of diffusion latents. It trains a lightweight flow-based generator (≈10⁴ parameters) initialized from Tree-Ring, enabling fast embedding, strong robustness under compound attacks, and post-hoc control over a quality–robustness trade-off without retraining. Experiments on MS-COCO, DiffusionDB, and WikiArt show improved robustness over several prior watermarking methods.

**Reviewer Concerns:**

Strengths (consensus)
	Strong engineering and evaluation: extensive robustness experiments, including compound and generative attacks.
	Practical design: no per-image iterative optimization, relatively efficient deployment, and adjustable robustness/quality setting.
	Robustness: consistently strong performance under regeneration and compound perturbations, which several reviewers identify as the paper’s main strength.

Main Concerns (driving low scores)
	Limited novelty: multiple reviewers argue the core idea—latent Fourier-domain watermarking with Tree-Ring-style initialization—largely refines existing methods (Tree-Ring, ZoDiac) rather than introducing a fundamentally new watermarking mechanism.
	Dependence on Tree-Ring: results degrade significantly without Tree-Ring initialization, suggesting WaterFlow is more an enhancement than an independent method.
	Quality and efficiency claims are not fully convincing:
	SSIM/PSNR are sometimes lower than baselines.
	Latency is insufficiently contextualized; comparisons to commercial or CPU-based systems are missing.
	Runtime comparisons vs. ZoDiac may be misleading given per-image adaptation of the learned watermark.
	Clarity issues: several methodological details (FFT assumptions, losses, symbols) are insufficiently explained, harming accessibility.

**Reviewer Scores:**

W4xZ: 2, Conf 5 — Reject; high confidence, strong novelty and motivation concerns.
MrTA: 4, Conf 4 — Marginally below; robustness promising but evaluation/claims weak.
ASCj: 4, Conf 4 — Marginally below; engineering-heavy, limited conceptual novelty.
EkDk: 4, Conf 4 — Marginally below; strong robustness but unclear contribution vs Tree-Ring.

---

### Decision · Program_Chairs · 2026-01-26

Reject